# Mapping fast DNA polymerase exchange during replication

Longfu Xu [1], Matthew T. J. Halma[1] & Gijs J. L. Wuite [1] ✉

Despite extensive studies on DNA replication, the exchange mechanisms of DNA polymerase during replication remain unclear. Existing models propose that this exchange is facilitated by protein partners like helicase. Here we present data, employing a combination of mechanical DNA manipulation and single fluorescent protein observation, that reveal DNA polymerase undergoing rapid and autonomous exchange during replication not coordinated by other proteins. The DNA polymerase shows fast unbinding and rebinding dynamics, displaying a preference for either exonuclease or polymerase activity, or pausing events, during each brief binding event. We also observed a 'memory effect' in DNA polymerase rebinding, i.e., the enzyme tends to preserve its prior activity upon reassociation. This effect, potentially linked to the ssDNA/dsDNA junction's conformation, might play a role in regulating binding preference enabling high processivity amidst rapid protein exchange. Taken together, our findings support an autonomous replication model that includes rapid protein exchange, burst of activity, and a 'memory effect' while moving processively forward.

The replication of duplex DNA, orchestrated by an ensemble of enzymes and proteins constituting the replisome, stands as an example of highly refined biological machinery. Traditional models have portrayed the leading- and lagging-strand polymerases as well-coordinated and intimately associated with the replisome, synthesizing DNA in a continuous manner[1–3]. However, replisomes in cellular environments encounter numerous challenges, including transcription complexes, DNA lesions, and constant nucleotide mismatches, which potentially disrupt the stability and defined structure of the replisome[4,5]. Contrary to that conventional model, recent evidence from single-molecule studies suggests a more dynamic scenario in viral T7 replisomes[5–7], bacteria[8–14], and eukaryotes[15,16]. These findings reveal that the replisome is not a stable entity, but rather highly dynamic, with components rapidly exchanging. In the T7 DNA replication system, the helicase and polymerase function in a coordinated manner, ensuring efficient leading- and lagging-strand synthesis. Investigations on DNA polymerase have demonstrated that polymerases are recruited to the DNA via interactions with T7 gp4, and that these interactions serve to increase local polymerase concentration and mediate polymerase exchange[17,18].

DNA polymerase is renowned for its dual functionalities of polymerization and exonucleolysis, with the latter responsible for proofreading[19]. A distinct exonucleolytic domain enables proofreading by providing an alternate binding site[20,21]. The transition from replication to proofreading is hypothesized to be governed by a conformational switch within the DNA polymerase, which may result in disengagement from the helicase due to opposing movement directions on the DNA. Despite the plausible model of DNA polymerase exchange coordinated by partner proteins, the precise mechanism by which DNA polymerase balances mismatch correction and helicase interaction remains enigmatic. Intriguingly, a recent single-molecule observation of live *E. coli* replisomes discovered that polymerases within the replisome function independently and discontinuously, without coordination[22].

To gain a deeper understanding of the replisome's exchange dynamics, it is essential to scrutinize these exchange processes within a well-controlled system. This demands experimental techniques that can detect critical intermediates and monitor their emergence and disappearance in real-time. Addressing this requirement, we develop a

[1]Department of Physics and Astronomy and LaserLab, Vrije Universiteit Amsterdam, De Boelelaan 1081, 1081 HV Amsterdam, The Netherlands.
✉e-mail: g.j.l.wuite@vu.nl

state-of-the-art correlative optical tweezers-fluorescence microscopy capable of real-time tracking of DNA polymerase intermediate transitions, building upon our earlier assays[19,23,24]. We demonstrate the application of this current assay in determining the mechanism of T7 polymerase exchange in vitro. T7 DNA polymerase comprises a stable complex of gene 5 protein (gp5) and thioredoxin, a processivity factor produced by the *E. coli* host[25]. The bacteriophage T7 DNA polymerase, sharing mechanistic pathway and structural similarities with higher organisms, serves as an ideal model system for investigating replication dynamics and deciphering molecular dynamics in real-time[25,26].

In this study, we leverage a synergistic combination of mechanical manipulation of DNA and observation of single fluorescent proteins to probe the T7 DNA polymerase dynamics during replication, independently of other partner proteins. The correlation of distinct single-molecule datasets grants access to kinetic activity and binding dynamics of the replicative polymerase. This, in turn, reveals that DNA polymerase molecules rapidly unbind and rebind to DNA, with each molecule predominately performing only one catalytic function (either polymerization or exonucleolysis or pause) during each binding event. Moreover, the rebinding of DNA polymerase during replication exhibits a memory effect, i.e., the catalytic activity is preserved over a longer time than the actual binding time of a single protein,

which might explain the chemically continuous yet kinetically discontinuous nature of replication. We also capture diffusive DNA polymerase on dsDNA which potentially aids in the exchange of DNA polymerase. Our findings elucidate a relationship between DNA polymerase activity pauses, unbinding events, and self-generated roadblocks, suggesting that these elements impact each other in a complex manner. We discuss these results within the framework of a decentralized replication model that connects these findings with faithful and apparent coordinated replication.

## Results

### Dynamic exchange of DNA polymerase captured by correlative optical tweezers-fluorescence microscopy

In this study, we employed correlative optical tweezers-fluorescence microscopy to investigate the exchange of T7 DNA polymerase at the replication junction. Using dual-trap optical tweezers, we tether single biotinylated pKYB1 DNA constructs (8393 bp) between beads, while a confocal microscope simultaneously records fluorescence from labeled DNA polymerase (Fig. 1A). Using an incomplete labeling strategy (~60% labeling ratio) to allow visualization of protein exchange, we developed a method to effectively track protein dynamics during replication (Methods). Figure 1B showcases an example force-distance

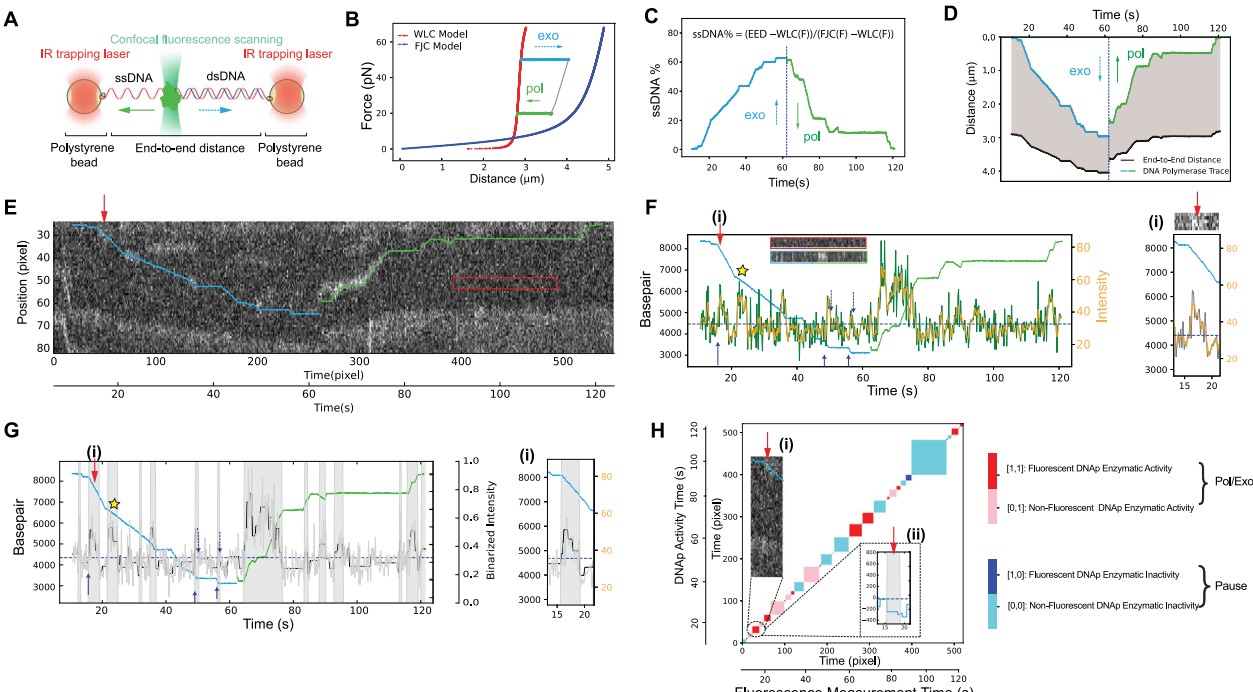

**Fig. 1 | Dynamic exchange of DNA polymerase captured by correlative tweezer fluorescence microscopy. A** Schematic of the correlative tweezers-fluorescence microscopy system, using dual optical tweezers trapping biotinylated pKYB1 DNA constructs between microspheres and a confocal microscope capturing fluorescence signals from labeled DNA polymerase. **B** Representative force-distance curve displaying force-dependent polymerization (pol, green) and exonucleolysis (exo, blue) activity of T7 DNA polymerase, fit with WLC (red) and FJC (purple) models, respectively. **C** Force-dependent T7 DNA polymerase activity illustrating exonucleolytic (blue) and polymerization (green) processes, via the change in end-to-end distance (EED) between trapped beads (panel **B**). ssDNA portion calculated assuming ssDNA follows FJC, dsDNA follows WLC. Purple dashed line: force transition. **D** Kymograph of ssDNA/dsDNA junction movement (blue and green line) over time, derived from DNA polymerase catalytic activity and the EED (solid black line), using Eqs. (3) or (4) (Methods). **E** Correlation of polymerase activity (panel **D**) and fluorescence signal (Supplementary Fig. 2A), overlaid by aligning local maximum intensities (panel **D**, Method). Note, that the top and bottom high-intensity

bands are from bead signals, red box for background subtraction. **F** (Left) Fluorescence signal of DNA polymerase at the ssDNA/dsDNA junction (filtered signal in yellow, raw data in gray, Methods). Solid blue (exo) and green (pol) lines represent force-dependent DNA polymerase activity. Note, horizontal dashed line: background threshold; purple solid up arrows: DNA polymerase binding events; purple dashed down arrows: dissociation events; yellow star: rate change suggesting exchange. (Right) Zoomed kymograph event (red arrow in panel **E**). **G** (Left) Correlation between DNA polymerase activity and its binding dynamics. The fluorescence signal (panel **F**) is binarized. Shaded area: segments with detected fluorescent protein. (Right) Zoomed event by the red arrow (**i**). In total, 36 individual DNA molecules were analyzed, yielding 177 segments for Exo events and 202 for Pol events. A gallery of 9 unique DNA polymerase traces is presented in Supplementary Fig. 1. **H** Correlation heatmap of binarized polymerase activity and binarized fluorescence intensity over time. *Inset*: zoomed polymerase trajectory (**i**, panel **E**); and the corresponding burst activity (**ii**, Supplementary Fig. 2B). Source data are provided as a Source Data file.

trace of the known force-dependent exonucleolytic and polymerization processes by T7 DNA polymerase, indicated by blue dashed and green solid arrows, respectively. DNA polymerase activity is measured by the change in base pairs and thus end-to-end distance (EED) of the DNA construct, as determined from the distance between the two optically trapped beads. We calculated the ssDNA percentage (Fig. 1C) using Eq. (1)[19] and the dsDNA percentage using Eq. (2) (see Methods), following the FJC model for ssDNA and the extensible WLC model for dsDNA elasticity[27,28]. The calculated DNA kymograph in Fig. 1D shows how this measured DNA length change would translate into ssDNA/dsDNA junction's movement using Eq. (3) (see Methods),

$$\text{Junction position} = \frac{\text{ssDNA\%} * \text{FJC}(F)}{\text{ssDNA\%} * \text{FJC}(F) + \text{dsDNA\%} * \text{WLC}(F)} * \text{EED} \quad (3)$$

This movement arises due to the catalytic activity of DNA polymerase and can be tracked using our correlative optical tweezers-fluorescence microscopy with fluorescently labeled DNA polymerase. By scanning the fluorescence along the DNA template over time, we can indeed directly visualize the movement of DNA polymerase on DNA and generate a 2D fluorescence kymograph (Fig. 1E). To streamline data interpretation, we aligned the calculated DNA kymograph (Fig. 1D) with the fluorescence kymograph of the labeled protein (Fig. 1E). An optimal overlap between the calculated ssDNA/dsDNA junction and the fluorescent trajectory was ensured by searching for a local maximum fluorescent intensity at the ssDNA/dsDNA junction (Methods). The alignment permits visual tracking of the DNA polymerase's movement along the junction in real-time. This holds true even when faced with intermittent disappearance of the fluorescence.

Using this method, we monitored DNA polymerase exchange in real-time. We extracted fluorescence intensity at the ss/ds DNA junction over time using a five-pixel width box (each pixel size of 75 nm), analyzing only the green channel to minimize crosstalk. By applying a threshold of 3 to 5 standard deviations above the background noise, we effectively filtered out noise from freely diffusing labeled proteins (Fig. 1F). With our method we can detect real-time binding events (some illustrated by purple solid up-arrows) and dissociation events (some illustrated by purple dashed down-arrows) of DNA polymerase molecules from the DNA template. Some instances of a change in DNA polymerase rate due to protein exchange are indicated with the yellow stars (Fig. 1F). Further analysis of the fluorescence signal along the ssDNA/dsDNA junction enabled the identification of individual binding and unbinding events of the DNA polymerase. Next, a step-fitting algorithm was employed, with a ~4 standard deviation threshold applied to account for background noise, to binarize the fluorescence signal. The thus obtained signal allows correlating the burst activity of DNA polymerase with its binding and unbinding events at the replication junction (Fig. 1G). This approach revealed that fluorescence signal intensity changes correspond to DNA polymerase's binding, unbinding, and catalytic activity in base pair incorporation and removal.

Next, we extracted all the segments with fluorescent protein binding. We defined single-type burst segments by a constant fluorescence signal with a constant protein reaction rate, and transitional-type segments by a constant fluorescence signal but with a change in protein rate. We observed a total of 177 exonucleolysis and 202 polymerization segments out of 36 different DNA molecules. A gallery of nine unique DNA polymerase traces is presented in Supplementary Fig. 1. Representatively, the right panels of Fig. 1F–i, G–i illustrate a typical segment that occurs at the red arrow.

We also probed the relation between protein dissociation and subsequent pausing events. Utilizing a step-fitting algorithm, we isolated discrete DNA polymerase bursts of activities and synchronized them with the binarized fluorescence signals (Supplementary Fig. 2B).

Considering the known rates of T7 DNA polymerase's exonuclease (~100 bp/s) and polymerase (~200 bp/s) activities[24], pauses were assumed to occur when the measured catalytic rate is 10 time slower than the functional enzymatic activity. Specifically, this occurred below 10 bp/s for forces greater than 45 pN, and below 20 bp/s for forces less than 20 pN. This delineates the enzymatic reaction detection threshold of our instrument. Notably, these pausing events frequently coincided with the dissociation of the fluorescent polymerase (examples are seen at the purple dashed down-arrows).

To perform correlation analysis between force and fluorescence data, we transformed both the DNA polymerase activity data and the fluorescence intensity data into binary form, relative to the pre-set noise threshold. This binary transformation allows us to create a correlation matrix, using the following set of binary combinations: [1,1] represents fluorescent, active DNA polymerase; [1,0] denotes fluorescent, pausing DNA polymerase; [0,1] signifies non-fluorescent, active DNA polymerase; and [0,0] indicates non-fluorescent, pausing. The resultant correlation matrix is plotted in Fig. 1H. The correlation matrix of DNA polymerase activity and fluorescence intensity separated for exonuclease and polymerase activity is presented in Supplementary Fig. 2C. These matrices clearly reveal rapid activity switching during replication and serve as a powerful tool for visualizing key information derived from a complex kymograph (See Supplementary Fig. 1 for a selection of correlation matrixes). To highlight an instance of rapid DNA polymerase activity switching, we include a magnified correlated trace (top left, Fig. 1E-i) focusing on events within the dashed circle, along with a synchronized burst activity view aligned with binarized fluorescence intensity (bottom right, Fig. 1E-ii). Furthermore, our correlation analysis revealed that for the measured DNA molecules, the sum of the fluorescent activity [1,1] and pausing [1,0] events reached a value of 483, while non-fluorescent activity [0,1] and pauses [0,0] accumulated a total of 733. Consequently, fluorescent events comprise ~40% of all observed events. This percentage is lower than the ~60% labeling ratio (see Methods) and presumably arises from pause events due to protein dissociation which are scored as non-fluorescent events.

## Mapping burst activity by DNA polymerase

Our analysis of the identified segments revealed a predominance of single-type burst segments, with typically one protein bound at the junction (163 out of 177 exo events, and 199 out of 202 pol events from 36 discrete DNA molecules). Figure 2A illustrates an example of a single-type burst event, distinguished by both a stable fluorescence signal and a constant protein rate. This suggests that the protein engages in only a single-catalytic activity, such as exonucleolysis, polymerization, or pausing; in this particular case, the fluorescent polymerase performs exonucleolysis and its activity ceases after the fluorescence signal disappears (shaded area in Fig. 2A), most likely due to dissociation. In contrast, Fig. 2B depicts a transitional-type burst event, where the protein performs more than one activity before dissociating from the DNA molecule. In this example, after exonucleolysis the protein pauses, yet the fluorescence remains for some more time, indicating the protein enters a pause (shaded area in Fig. 2B). We mapped the distribution of these events (Fig. 2C, D). For exo, we note that single-catalytic events constitute ~90% of the total, and for pol, we see that single-type burst events happen in 98% of the cases. Despite the difference in template tension for exo and pol events, both distributions show that DNA polymerase preferentially engages in a single activity during one binding event. We observed that these burst events, typically lasting around 1 s, are significantly shorter than the photobleaching duration of ~13 s (Supplementary Fig. 3D, E). This notable time discrepancy ensures that our data interpretation isn't significantly influenced by photobleaching, as the DNA polymerase likely dissociates or exchanges with another one from the solution in this timeframe.

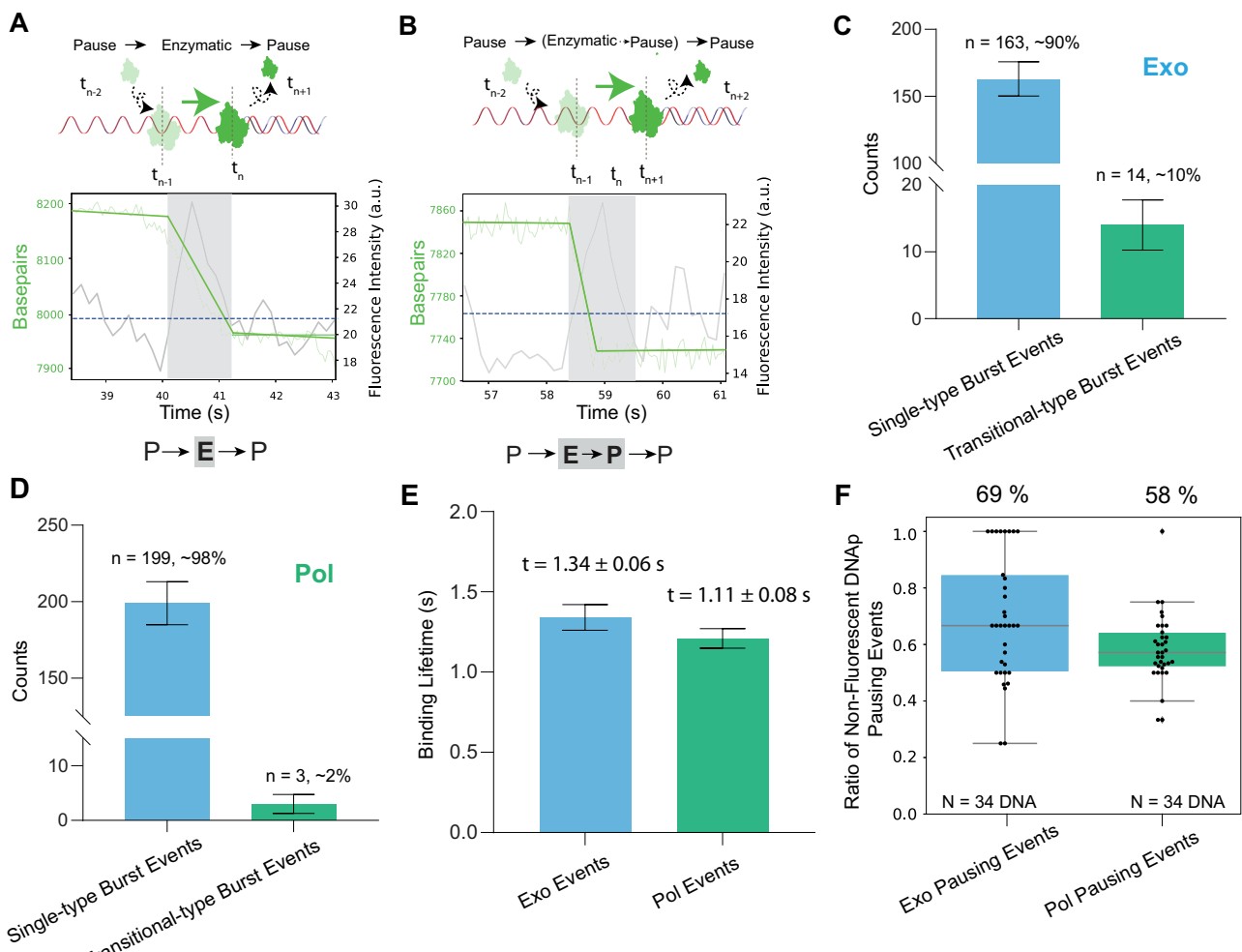

**Fig. 2 | Mapping burst activity by DNA polymerase. A** Top: Schematic of a single-type burst event. Bottom: Representative trace showing constant fluorescence and protein rate during activities like exonucleolysis, polymerization, or pausing. The shaded area indicates fluorescent protein presence. **B** Top: Schematic of a transitional-type burst event. Bottom: Representative trace showing constant fluorescence with varying protein rates, as the protein undergoes exonucleolysis, polymerization, and pausing before DNA unbinding. **C** Distribution of burst events during exonucleolysis, with single-type bursts making up ~90% and transitional-type bursts accounting for ~10%. Sample size: $n = 177$ exo segments from $N = 36$ DNA molecules. **D** Distribution of burst events during polymerization, with single-type bursts dominating at ~98%, while transitional-type bursts constituting ~2%. $n = 202$ pol segments from $N = 36$ DNA molecules. Error bars for (**C**, **D**): square root of the number of observations. **E** Bound lifetime of T7 DNA polymerase under exonucleolysis (50 pN) and polymerization (20 pN) conditions. Bound lifetime is defined as fluorescence duration within a segment ($n = 177$ exo segments, 202 pol segments). Mean lifetime obtained by fitting mono-exponential decay curve, adjusted for photobleaching. Under exo, the average lifetime is $1.34 \pm 0.06$ s, and under pol, $1.11 \pm 0.08$ s. Note: The estimated average bound lifetime may exceed the actual duration due to undetectable rapid exchange events limited by temporal resolution. Note, for binding duration analysis, events near excessively bright trapped beads (Supplementary Fig. 1) were not excluded. Error bars: errors from mono-exponential fit. **F** Proportion of non-fluorescent DNA polymerase pausing events under exonuclease and polymerase conditions (Methods). Box plots show the median (central line), quartiles (box boundaries), and full data range (whiskers). The analysis incorporated a total of 275 exonuclease and 425 replicative pausing events from $N = 34$ DNA molecules (no pausing events were observed for 2 DNA), where non-fluorescent pauses accounted for proportions with a median of 0.69 (25th percentile: 0.63, 75th percentile: 0.77) and 0.58 (25th percentile: 0.54, 75th percentile: 0.63) for exonuclease and replicative conditions, respectively (mean ± SEM: $0.69 \pm 0.04$ and $0.58 \pm 0.02$). Source data are provided as a Source Data file.

To unravel the T7 DNA polymerase dynamics, we evaluated the bound lifetime under predominantly exo and pol conditions, i.e., at high and low tension (Fig. 2E). The bound lifetime was quantified as the duration of a single fluorescence signal. Using data from 177 exo segments and 202 pol segments, we derived the mean binding lifetime by fitting the data to a mono-exponential decay, introducing adjustments to account for potential fluorophore photobleaching. Our analysis yielded a mean lifetime for exo events of $t = 1.34 \pm 0.06$ s, and for pol events, $t = 1.11 \pm 0.08$ s. Notably, these estimated mean binding durations likely represent upper limits, considering the inherent temporal resolution constraints of our instrument in detecting rapid exchange events.

Next, we computed the average rate during each visible i.e., labeled protein, binding event as depicted in Supplementary Fig. 2D. While there was substantial variation in the burst activity distribution, the force-dependent nature of the binding activity was still clearly apparent. At 50 pN tension, the predominance of exonuclease activity is evident, with a rate of $-62 \pm 4$ bp/s (mean ± SEM). In contrast, at 20 pN, the polymerase activity is dominant exhibiting an average rate of $32 \pm 7$ bp/s. This force-dependent activity aligns with prior studies[19,24], albeit with slightly reduced average rates possibly due to the inclusion of pausing events in our mean rate calculations.

Furthermore, we determined the distribution of non-fluorescent pausing events for each individual DNA molecules (Fig. 2F). This

distribution, quantified as the fraction of the [0,0] events relative to the total of [0,0] and [1,0] events, encapsulates the degree of non-fluorescent pausing events among all inactive states. This data reveals that the percentage of non-fluorescent DNA polymerase pausing is $69 \pm 4\%$ and $58 \pm 2\%$ for exonuclease and replicative pausing, respectively (mean ± SEM). Given that 40% of the protein is unlabeled while ~60% of the pauses are non-fluorescent, this suggests that a significant correlation exists between pausing and protein disassociation.

### Memory effect of the binding state of DNA polymerase at the replication junction

The short binding life-time we find at the ssDNA/dsDNA junction (Fig. 2E) contrast with the high processivity of DNA polymerase observed during in vivo replication[29,30]. Hence, we investigated whether the binding state of DNA polymerase at the ssDNA/dsDNA junction influences its subsequent activity and whether some kind of memory effect is present that would explain the reported processivity. To elucidate this potential memory effect, we carried out a detailed examination as presented in Fig. 3. Figure 3A shows a typical event that illustrates such a potential memory effect. The top kymograph displays a distinct protein event bound at the ssDNA/dsDNA junction (marked by the yellow line, as described in Fig. 1), with a close-up view of this specific protein trajectory shown in the middle panel. The lower panel offers a magnified view of the correlation between DNA polymerase activity and fluorescence bursts within the fluorescence segments. The shaded area signifies the presence of a fluorescent protein, whose intensity surpasses the background threshold (in this case 3 sigma was used, see Methods, Supplementary Fig. 3A–C). DNA polymerase activity can be tracked by observing changes in base pairs. In this case, these changes reveal that the DNA polymerase is actively performing exonucleolysis, removing bases and thus decreasing the total numbers of base pairs. We've marked the potential moment of rapid exchange—a transition in DNA polymerase activity as signified by a slight change in the rate—with red arrows (Fig. 3A, bottom). Whenever there is a change in rate, we also observe a significant fluctuation in the florescent signal which could correspond to a rapid protein exchange (see Supplementary Fig. 3A–C for a comparison of the fluorescence signal of a static DNA polymerase). These two signals together point toward rapid exchange while one catalytic activity persists.

However, to discern if there is an actual memory effect associated with DNA polymerase binding instead of just a chance process, we mapped all the single-type burst events. Focusing on three primary protein states: polymerase (Pol), exonuclease (Exo), and pause (P); for simplicity, we represented both Pol and Exo states as enzymatic activity (E). Considering two potential states (E and P) at three different positions (pre-, during, and post-fluorescent protein binding), we identified eight distinct configurations of state switching within a segment (Fig. 3B). The distribution between these 8 configurations during exonucleolytic activity (at 50 pN) is shown in Fig. 3C. Notably, the E→E→E sequences were most prevalent, accounting for 34% of events, closely followed by P→P→P sequences, comprising 23%. This outcome indicates that the enzymatic state after an exchange is likely dependent on the preceding state, thus pointing towards the existence of a memory effect. In Fig. 3D, we present the distribution of the eight configurations during polymerase activity (at 20 pN). Despite the distribution diverging from that observed at 50 pN (during exo events), the two most prevalent sequences remained the same: P→P→P and E→E→E. This consistency suggests that this memory effect could result from the ssDNA/dsDNA junction conformation, which does govern the binding interaction with the protein, in agreement with our finding that DNA polymerase activity is force dependent (Fig. 2F).

### Dynamics of DNA polymerase at the replication junction influenced by ssDNA and dsDNA

DNA polymerase wants to bind at the ssDNA/dsDNA junction, facilitating exonuclease or polymerization activities. However, our observations also revealed diffusing DNA polymerases on the dsDNA (5 traces out of 36 DNA molecules). An example trace (Fig. 4A right) depicts the calculated ssDNA/dsDNA junction movement over time (green line) and showing a diffusing DNA polymerase on the dsDNA before potential entry into the replication junction. We also observe DNA polymerase binding to ssDNA, especially at a higher concentration (>30 nM). Yet these protein bindings appear stationary (see for example, Fig. 4B) and seem to be able to obstruct DNA polymerase movement (11 traces from 3 distinct DNA molecules) (Fig. 4B, right panel). To further investigate these binding events, we computed the corresponding diffusion constants using Mean Square Displacement (MSD) (Fig. 4C). The analysis revealed a stark contrast between the diffusion constants on ssDNA and dsDNA. The average diffusion constant for DNA polymerase on ssDNA was measured to be $0.003 \pm 0.002 \, \mu m^2/s$ (mean ± SEM), and $0.02 \pm 0.003 \, \mu m^2/s$ on dsDNA. The diffusion on ssDNA is essentially negligible, while the diffusion value on dsDNA is somewhat lower than previously reported[31], possibly due to tension applied on the DNA template. The diffusion of DNA polymerase on dsDNA could be a searching mechanism[31] employed by DNA polymerase to find the replication junction. In contrast, the static binding of DNA polymerase on ssDNA may lead to the formation of 'roadblocks.' (Fig. 4B). In fact, the average lifetime of DNA polymerase on ssDNA is $6.5 \pm 0.4 \, s$ (mean ± SEM; $N = 335$; Fig. 4D, E) which is five times longer than DNA polymerase on the ssDNA/dsDNA junction. This extended binding duration on the ssDNA and the observed static behaviors support the notion that it can indeed act as a temporary impediment to replication.

## Discussion

Combining mechanical measurement of DNA with the observation of individual fluorescent proteins, enabled us to track the real-time dynamics of DNA polymerase during replication. By observing individual DNA polymerase actions under conditions that promote replication and proofreading without contact with other protein partners, we demonstrate that DNA polymerases undergo rapid exchange at the ss/dsDNA junction in an independent and stochastic manner. DNA replication has often been viewed as a continuous leading-strand synthesis process with very high processivity[1–3]. However, recent in vivo and in vitro studies have suggested that individual DNA polymerase exchange may occur due to helicase coordination[5,10,17,32]. Our findings show rapid exchange occurring independent of a helicase. This finding is also consistent with previous reports[19,24] and findings for other systems such as the bacteriophage T4 replication system[33] and the *E. coli* system[34]. We also observe that the proteins perform distinct roles before exchanging with other proteins in the immediate environment. The rapid exchange of DNA polymerases, coupled with a memory of the binding state, allows for an apparent high processivity. Furthermore, we observe DNA polymerases diffusing on dsDNA which might facilitate rapid exchange at the ss/dsDNA junction. While DNA polymerases that pause at the replication junction, dissociate from the junction, or bind to ssDNA ahead of the advancing replication junction, can instigate pausing events. These findings illustrate the diverse dynamics of DNA polymerase at the replication junction and underscore the influence of ssDNA and dsDNA on replication dynamics.

DNA polymerase has been recognized for its integral role in maintaining replicative fidelity by incorporating an intrinsic proof-reading activity[2,33]. Ensemble measurements and single-molecule investigations have shaped the model where DNA polymerase is anticipated to discern misincorporations and, with remarkable precision[2,33,35], switch to the exonucleolytic catalytic site, despite the energy cost of conformational changes for ~60-Å distance within the

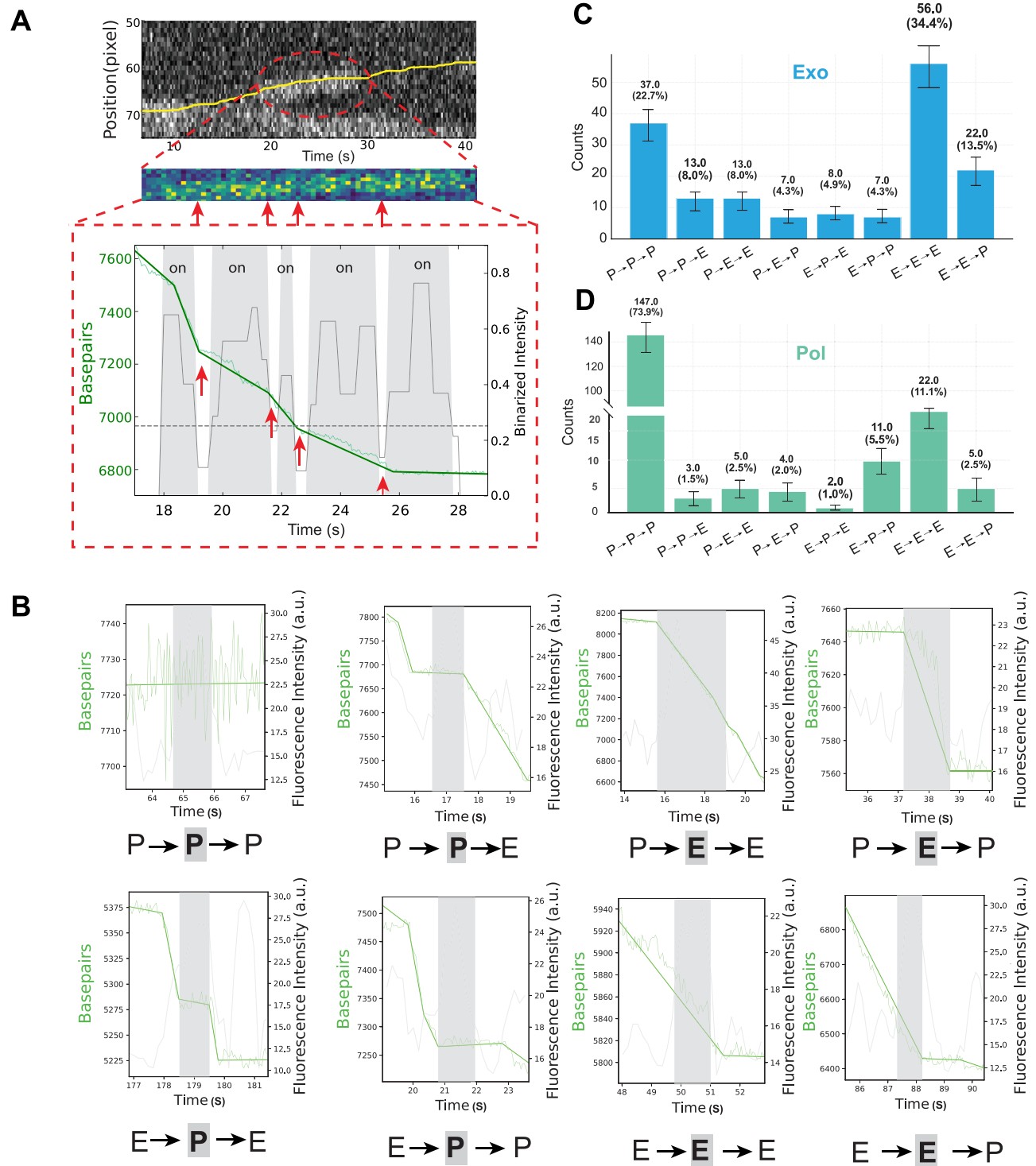

protein[36,37]. This has raised questions about how misincorporation signals T7 DNA polymerase to transit to exonuclease activity[24,36,38]. Our data reveal that individual DNA polymerase molecules typically undertake a single function during their short-lived binding events. With a rate of incorrect nucleotide incorporation into the growing strand of roughly one per $10^6$ synthesized bases[2,33,35], achieving high fidelity necessitates the consistent correction of misincorporated nucleotides by DNA polymerase. Importantly, this proofreading process could induce frequent decoupling events with the advancing helicase, subsequently leading to pauses in overall replisome activity while awaiting the lagging proofreading DNA polymerase. Our study

proposes a model wherein the conformational states associated with DNA polymerase's catalytic activity (polymerization, exonuclease activity, or paused) are thermally distributed in solution. The ss/dsDNA junction conformation acts as a selector, determining which DNA polymerase conformation can successfully bind. Once bound, DNA polymerase typically retains its conformation and performs a single function until detachment. In cases of misincorporation, the replicative DNA polymerase detaches, allowing another polymerase to bind at the junction with its exonuclease site ready. Consequently, the catalytic direction of the protein appears to be governed by the ss/dsDNA junction conformation, exhibiting force-dependent binding activity,

**Fig. 3 | Rapid exchange events and the potential memory effect of polymerase binding state. A** Representative rapid exchange event. (Top) Kymograph displaying a bright protein event bound at the ssDNA/dsDNA junction (yellow line), with the DNA junction position calculated as described in Fig. 1. (Middle) A zoom-in view of the specific protein trajectory, encircled by a dashed red ellipse. (Bottom) A zoom-in view of the correlation between DNAp activity and fluorescence bursts within the fluorescence segments. The shaded region signifies the presence of a fluorescent protein (exceeding the background threshold, indicated by a horizontal dashed line) bound at the junction. The red arrow indicates the potential timestamp of rapid exchange. The high-intensity bands at the bottom of the kymographs are due to fluorescence signals from the beads used in our optical tweezer's setup. **B** Eight distinct configurations of state switching in single-type burst segments, focusing on three protein states: polymerase (Pol), exonuclease (Exo), and pause (P). Pol and Exo states are simplified as Enzymatic activity (E).

Given two possibilities (E and P) in three positions (pre-, during, and post-fluorescent protein binding), there are eight potential configurations of state switching. **C** Distribution of the eight configurations in exonuclease events under 50 pN, with E→E→E sequences being the most prevalent (34.4%), followed by P→P→P sequences (22.7%). Transitions appear to be dependent on preceding transitions, suggesting a memory effect. Sample size: $n = 163$ single-type burst events out of 177 exo segments (measured from $N = 36$ DNA molecules). **D** Distribution of the eight configurations in polymerase events under 20 pN, pointing to a force-dependent memory effect and ssDNA/dsDNA junction conformation change. Despite the variation in the distribution of the eight configurations, the two most common sequences are P→P→P and E→E→E, hinting at a possible memory effect. Sample size: $n = 199$ single-type burst events out of 202 pol segments (measured from $N = 36$ DNA molecules). Error bars for (**C**, **D**): square root of the number of observations. Source data are provided as a Source Data file.

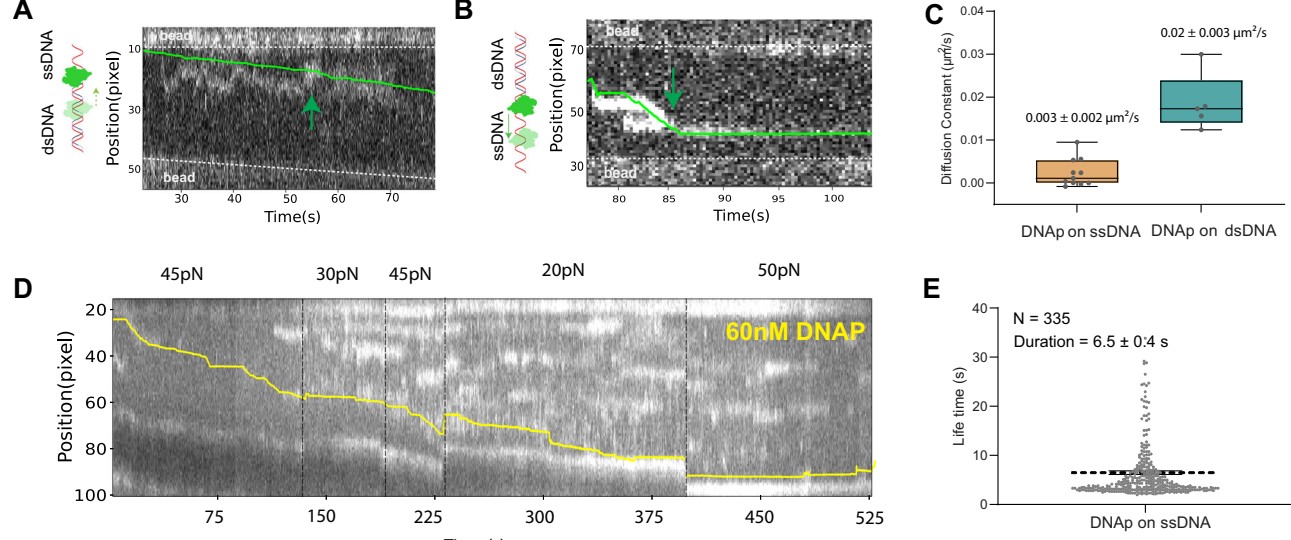

**Fig. 4 | DNA polymerase dynamics on ssDNA and dsDNA regimes. A** (Left) Schematic of a diffusive DNA polymerase (light green) on dsDNA, searching for the replication junction and encountering an exonuclease polymerase (green). (Right) Kymograph of fluorescently labeled T7 polymerase diffusing on dsDNA; green line indicates ssDNA/dsDNA junction. The green arrow highlights a diffusive polymerase binding to the junction from dsDNA. The high-intensity bands at the top and bottom of the kymographs are due to fluorescence signals from the beads; this applies to all the kymographs in this figure. **B** Schematic of a replicative polymerase paused at the junction due to a stalled polymerase on the ssDNA template. (Right) Kymograph showing a replicative polymerase encountering an extra polymerase bound on ssDNA, resulting in replication cessation. **C** Comparison of polymerase diffusion constants on ssDNA and dsDNA using mean square displacement (MSD), revealing more diffusive behavior on dsDNA. Average diffusion constant on ssDNA: $0.003 \pm 0.002$ μm²/s ($n = 11$ traces out of 3 distinct ssDNA molecules); on dsDNA:

$0.02 \pm 0.003$ μm²/s ($n = 5$ traces out of 36 dsDNA molecules). Values: mean ± SEM. Note that the box plots represent the distribution of diffusion constants, with the horizontal line indicating the median, box boundaries as the 25th and 75th percentiles, whiskers extending to 1.5 times the interquartile range. **D** Kymograph of fluorescent polymerase binding to ssDNA under varying tensions. Polymerase binds at ssDNA/dsDNA junction (yellow line), increasing ssDNA fraction as more polymerase binds between 20–50 pN. The yellow line separates the kymograph into ssDNA (above the yellow line), dsDNA (below the yellow line), and junction regions (coincident with the yellow line) (Methods). Note that the polymerase concentration is set to be 60 nM in the absence of dNTPs, aiming to generate a long ssDNA region. **E** Distribution of polymerase lifetime on ssDNA at 20 pN. Fluorescent DNA polymerase trajectories were extracted, and binding times were averaged. Sample size: $n = 335$ traces from $N = 7$ distinct DNA molecules; an average lifetime of $6.5 \pm 0.4$ s (mean ± SEM). Source data are provided as a Source Data file.

consistent with prior findings[24,39]. Thus, it seems that error correction by DNA polymerase is a highly stochastic process, rather than a precise controlled mechanism. Such stochastic binding might represent a broader strategy among multifunctional molecular machines, favoring state change through unbinding and rebinding, as a more energetically efficient method than direct switching.

Two distinct pausing states for T7 DNA polymerase (long-pauses and short-pauses) have been reported[24,40]. The origin of these pauses remains unclear, with hypotheses suggesting polymerase stalls at difficult-to-replicate DNA sequences or roadblocks, and/or pauses resulting from the enzyme's uncertainty about which direction to move. Our single-molecule measurements unveil a positive correlation between protein dissociation and short pausing. This observation

corroborates earlier findings which indicate concentration-dependent short pausing at low concentrations[24]. DNA polymerase is also recognized for its affinity for ssDNA, a phenomenon substantiated by ensemble measurements[41]. We provide single-molecule data showing static binding of DNA polymerase on ssDNA, which conceivably gives rise to self-imposed roadblocks. These roadblocks, in turn, seem to trigger long pausing events.

Prior single-molecule replication studies normally face challenges in determining DNA polymerase exchange due to either the limitations of mechanical measurements or low spatiotemporal resolution in imaging measurements. Our method, correlating fluorescent protein trajectories with mechanical measurements, overcomes these limitations and enables real-time monitoring of both DNA polymerase

activity and position. This approach allows for the discrimination of various dynamic behaviors, tracking the origin, direction, and transition time for each event, thus providing a comprehensive understanding of the underlying processes. Nevertheless, this method is still limited by the temporal resolution of confocal microscopy, especially the scanning rate along the DNA template (-0.3 s/scan). This restriction could lead to missing brief exchange events happening quicker than the scanning rate. Despite this, our assay presented here holds promise for broad applicability in addressing structure-function questions involving multifunctional protein acting on DNA. Future refinements, such as enhanced spatial-temporal resolution and the incorporation of established single-molecule fluorescence techniques like FRET[42] and fluorescent anisotropy[43], will offer deeper structural insights and facilitate direct connections between conformational changes and the activity of a multifunctional protein.

## Methods

### DNA template construction and fluorescently labeled DNA polymerase preparation
In our study, we utilized a ssDNA/dsDNA construct mimicking part of the replication fork, composed of a single dsDNA template and primer, to explore DNA polymerase dynamics under controlled conditions. This construct, with biotin labels at both ends, was prepared following established protocols[44], see Supplementary Methods and Supplementary Fig. 4. Briefly, the pKYBI vector was restricted, and biotinylated dATP was added using a Klenow reaction. A 5'-biotinylated 29-mer was ligated to the 3' ends, generating a DNA construct with a 25 nt 5'-end overhang, enabling exonucleolysis to create an ss/dsDNA junction for polymerization. By switching between low and high tensions, repeated cycles of replication and exonucleolysis on a single DNA molecule were achieved.

Recombinant SNAP-tagged T7 DNA polymerase (-100 KDa) was expressed, purified, and labeled using established protocols with modifications[45,46]. Briefly, SNAP-tagged T7 DNA polymerase is genetically engineered by attaching a SNAP-tag at the N-terminus of gp5 gene. The resulting SNAP-tagged T7 DNA polymerase was overexpressed in *E. coli* Rosetta 2 cells, purified using Ni-NTA beads, and labeled with SNAP-Surface® 549. After labeling, the labeled polymerase was filtered to remove any unbound or excess fluorophores. The fluorescence labeling efficiency was -60%. Pure and labeled proteins were aliquoted, flash-frozen, and stored at −80 °C. Detailed protocols are provided in the Supplementary Methods. Characterization of the SNAP-tagged T7 DNA polymerase, including expression analysis, structural modeling, and functional activity test, is presented in Supplementary Figs. 5, 6, and 7, respectively.

### Single-molecule assays
We performed single-molecule experiments using a LUMICKS C-Trap instrument, integrating three-color confocal fluorescence microscopy, dual-trap optical tweezers, and a five-channel microfluidic flow cell. Biotinylated pKYB1 DNA constructs were tethered in situ between two 1.76 μm streptavidin-coated microspheres (Spherotech Inc) within the flow cell. The presence of a single DNA molecule was confirmed via a change in the F-x curve. All experiments were conducted at room temperature, with the flow turned off during data acquisition. Force data was collected at 20 Hz, while fluorescently labeled DNA polymerase was excited at 532 nm.

Before experiments, the multichannel laminar flow cell was cleaned with bleach and passivated overnight with 0.1% BSA (NEB) and 0.5% Pluronic (Sigma Aldrich) to prevent nonspecific enzyme binding. Experiments were typically performed in a standard measurement buffer containing 20 mM Tris-HCl pH 7.5, 100 mM NaCl, 3 mM MgCl$_2$, 1 mM DTT, and 0.02% BSA, unless otherwise stated. DNA polymerase concentration was set at 30 nM unless specified otherwise. The dNTPs (mix) concentration was set at 625 nM for all experimental conditions,

and we included a 2–4-fold molar excess of thioredoxin relative to the T7 DNA polymerase in our experiments[46,47], unless otherwise stated.

In our study, we employ a simplified replication system, focusing on DNA elongation on a template and primer, without the presence of other proteins like helicase. This approach allows us to specifically examine the intrinsic exchange dynamics of T7 DNA polymerase, independent of the complexities introduced by the entire replication machinery. DNA polymerase activity was assessed by monitoring DNA extension changes at a constant tension of 40–50 pN to digest -5 kbp (>50% of total length) dsDNA, creating a long ssDNA section as a polymerization template. Subsequently, the tension was reduced to 10–20 pN for polymerization activity measurement. Additional details regarding experimental procedures can be found in the figure legends, or in the main text.

In this study, we used a recombinant DNA polymerase with a SNAP-tag for protein labeling. Future research should consider direct labeling methods for the polymerase, although attempts to do so were not successful in our case.

The temporal resolution of our correlative optical tweezers-fluorescence microscopy setup is determined by the confocal scanning rate and the signal acquisition time required for reliable fluorescence detection. While this resolution does not allow for direct observation of single nucleotide incorporation events, it is optimized to capture the rapid exchange dynamics of DNA polymerase and polymerase activity patterns during replication, complementing higher-resolution techniques.

In our study, the signal-to-noise ratio (SNR) in kymographs is influenced by several factors, including the fluorescence intensity of the labeled proteins and the dynamic nature of polymerase interactions with DNA. We optimized the labeling of the T7 DNA polymerase to achieve a balance between detectable signal and preserving the native functionality of the enzyme.

### Data analysis
All data were obtained from C-trap in. tdms format and is analyzed using custom-written Python scripts.

**Determining ssDNA/dsDNA junction trajectory to generate DNA kymograph.** In our experiments, we used an optical tweezer system to manipulate template tension. Following earlier studies[18,19,24], T7 DNA polymerase shows a force-dependent behavior on DNA template, stemming from the mechanical forces affecting the structure of the nucleic acids. In this study, T7 DNA polymerase was set to perform exonuclease (exo) activity under 40–50 pN, generating partially ssDNA, and polymerization (pol) activity under 10−20 pN, synthesizing dsDNA. Consequently, the end-to-end distance, comprising both ssDNA and dsDNA portions, changed over time due to DNA polymerase's catalytic activity. Given the elasticity differences between ssDNA (described by an FJC model[28]) and dsDNA (described by an extensible WLC model[27]), the ssDNA percentage can be derived using the equation:

$$\text{ssDNA\%} = \frac{\text{EED} - \text{WLC}(F)}{\text{FJC}(F) - \text{WLC}(F)} \quad (1)$$

where EED is the end-to-end distance measured directly between two optically trapped beads. The dsDNA percentage can be calculated as

$$\text{dsDNA\%} = 1 - \text{sDNA\%} \quad (2)$$

The ssDNA/dsDNA junction position can be determined by assuming that ssDNA appears on the top half of the kymograph:

$$\text{Junction position} = \frac{\text{ssDNA\%} * \text{FJC}(F)}{\text{ssDNA\%} * \text{FJC}(F) + \text{dsDNA\%} * \text{WLC}(F)} * \text{EED} \quad (3)$$

If the dsDNA appears on the top half of the kymograph, the junction position can be derived as:

$$\text{Junction position} = \left(1 - \frac{\text{ssDNA\%} * \text{FJC}(F)}{\text{ssDNA\%} * \text{FJC}(F) + \text{dsDNA\%} * \text{WLC}(F)}\right) * \text{EED}$$

(4)

We can plot the junction movement alongside the end-to-end distance changes as a function of measurement time (Fig. 1A, D).

Notably, the applied forces (45–50 pN for exonuclease and 10–20 pN for polymerase activity), combined with our DNA tethering method and buffer conditions (100 mM NaCl and 3 mM MgCl$_2$), strongly support that the observed exonuclease activities are due to DNA polymerase action, instead of DNA peeling, which typically occurs under high tension (> 65 pN) with low ionic strength conditions[27,48].

**Correlating ssDNA/dsDNA junction movement with fluorescently labeled protein trajectory.** Alongside the distance-time curve, we recorded fluorescence trajectories of labeled proteins, generating a kymograph by scanning along the DNA molecule over time (scanning interval of 8 ms and pixel dwell time of 1–2 ms unless stated otherwise). The fluorescence trajectory represents the replicative or exonuclease DNA polymerase functioning at the ssDNA/dsDNA junction. The ssDNA/dsDNA junction movement, derived from the independent measurement of DNA length, is overlaid with the DNA polymerase trajectory in the kymograph to track DNA polymerase in real-time. Initially, the two independent measurements are manually overlapped, adjusting the x-offset (due to different starting times between the optical tweezer and fluorescence microscopy systems) and y-offset (due to different custom-defined positions in the region of interest from the wide-field windows in both systems). Subsequently, the fluorescence intensity along the ssDNA/dsDNA junction over time is extracted, focusing only on the green channel's fluorescence signal to minimize crosstalk. A box with a total width of five pixels around the junction line is employed to collect sufficient signals within the point-spread function, gathering photons in the box over time. The optimized overlap between the calculated ssDNA/dsDNA junction and the fluorescent trajectory is expected to yield maximum intensity. To find the optimized overlap, a search for local maximum intensity is performed by moving the box in the x- and y-direction within a five-pixel range, calculating the total photon value at each position (x,y). The x-offset and y-offset are then updated and used for further analysis.

**Discrimination of DNA polymerase binding positions: ssDNA, dsDNA, or ssDNA/dsDNA junction.** DNA polymerase transforms dsDNA into ssDNA by removing mismatched base pairs in a 3′ to 5′ direction and replicates ssDNA to dsDNA in a 5′ to 3′ direction. To discriminate between binding positions (ssDNA/dsDNA junction, dsDNA, or ssDNA), the ssDNA/dsDNA junction trajectory is correlated with the fluorescent protein trajectory. The fluorescence kymograph is virtually divided into three parts: ssDNA, dsDNA, and ssDNA/dsDNA junction. During proofreading activity, DNA polymerase moves toward the dsDNA region, causing dsDNA to shrink and ssDNA to expand. Conversely, during replication, the direction reverses. By analyzing DNA polymerase traces on ssDNA, dsDNA, and the ssDNA/dsDNA junction separately, their respective binding positions can be determined.

**Determination and mapping of single-type burst DNA polymerase segments.** After extracting the fluorescence signal at the ssDNA/dsDNA junction over time, we observed stepwise increases and decreases in fluorescence due to the binding and unbinding of fluorescent DNA polymerase. We used a previously established step-fitting algorithm[49] to identify these discrete steps. To mitigate background signal effects, we employed a box of equal size for background

extraction. The fluorescence signal was then binarized using a threshold set at three to five times the standard deviation of the signal, resulting in values of 0 (no detected fluorescent protein) or 1 (detected fluorescent protein). We calculated base pair changes over time as a function of the ssDNA portion of the total DNA length (8393 bp). By correlating fluorescence signal changes with base pair changes on the same time scale, we ensured coherence in our data analysis. Next, we focused on segments with a binarized intensity of 1, applying a default threshold filter to further refine our results. This filter exclusively included events with over ~3 adjacent fluorescent pixels, which corresponds to ~ 1 s, thereby ensuring the capture of significant binding events while minimizing noise. As a result, only pertinent events were considered in our analysis. We categorized the events into two types: single-type burst events, characterized by a constant fluorescent signal and protein rate, and transitional-type events, marked by a constant fluorescence signal but a variable protein rate. Our analysis encompassed 36 individual DNA molecules, yielding 177 segments for Exo events and 202 for Pol events.

**Determination of DNA polymerase activity.** The detailed method is based on previous publications with minor modifications[24]. In brief, we analyzed the base pair versus time traces to identify changing points, which mark shifts in the polymerase (pol) or exonuclease (exo) activity. Initially, we mitigated the noise in the traces using a Savitzky-Golay (SG) filter with a window size of 15. The first derivative of these filtered traces was then calculated, serving as the basis for identifying significant changes in the pol or exo trend. To detect these changes, we applied a step-detection method used in prior research[49]. This allowed us to mark the steps or change points in the base pair-time traces. The processivity, velocity, and duration are calculated for each segment.

One crucial aspect of our study involves the detection and characterization of pausing events, denoted by temporary halts in polymerase activity that yield steady base pair values over a significant period. The "noise threshold" is determined as the standard deviation of base pair-time traces in the absence of protein under the tensions under investigation, estimated by employing the median absolute deviation (MAD) of sequential data points and amplified by a constant factor (1.4826). This method facilitates a robust distinction between inherent fluctuations (noise) and genuine pausing events (signal). The "significant period" is deduced from the known exonuclease rate of T7 DNA polymerase (~100 bp/s) and polymerase rate (~200 bp/s)[24], indicative of the standard duration necessary for the polymerase to add a base pair under our experimental parameters. We define a pause as an event where the actual process duration is ten times greater than the expected time, which, for simplicity, is defined to occur at 10 bp/s under 45 or 50 pN and 20 bp/s under 20 or 10 pN.

**Calculation of binding lifetime of DNA polymerase.** In the single-molecule measurements, the bound duration can be directly quantified as the duration of the fluorescence signal within one segment (with binarized intensity of 1). Subsequently, the distribution of all binding events is plotted and corrected for the degree of fluorophore photobleaching. We then derived the mean binding lifetime by fitting the data to a mono-exponential decay.

**Quantification and correlation of DNA polymerase pausing events with fluorescence intensity over time.** The respective exonuclease (exo) and polymerase (pol) activities of DNA polymerase are quantified and correlated with binarized fluorescence signal intensity. A step-fitting algorithm is employed to delineate discrete DNA polymerase burst activities, which are then aligned with the binarized fluorescence signals (Supplementary Fig. 2B as an example).

To facilitate correlation analysis, the DNA polymerase activity is binarized relative to a pre-established noise threshold. This binarization is applied to both sets of data: DNA polymerase activity and

fluorescence intensity. This process allows for the construction of a correlation matrix, in which each matrix entry signifies a custom correlation value between the two datasets at a specified time point (in pixels). These correlation values are manually determined in line with the binarized protein activity and fluorescence intensity data, generating a set of binary combinations: [1,1], [1,0], [0,1], and [0,0]. Correlation values are color-coded: [1,1] is indicated in red representing fluorescent and active DNA polymerase; [0,1] is shown in pink representing non-fluorescent yet active DNA polymerase; [1,0] is depicted in blue, indicating fluorescent proteins during pausing events; [0,0] is represented in cyan, suggesting pausing events, likely triggered by protein dissociation. The diagonal of this heatmap, extending from the bottom left to the top right, illustrates the correlation values. Molecular events, categorized as consecutive instances of identical matrix values, are discerned, and their frequencies are enumerated.

Both exo and pol conditions undergo measurement of pausing events (See Supplementary Fig. 2C). The proportional distribution of non-fluorescent DNA polymerase pausing events is calculated as the number of [0,0] events divided by the sum of [0,0] and [1,0] clusters, thereby providing a metric for the extent of non-fluorescence pausing relative to overall non-active states. Notably, only clusters lasting over 3 pixels, which corresponds to ~1s, were included in the analysis. A total of 275 exonuclease pausing events and 425 replicative pausing events were observed from 34 distinct DNA molecules (no pausing events were observed for 2 DNA). Within these, non-fluorescent DNA polymerase pauses represented a proportion of $0.69 \pm 0.04$ (mean $\pm$ SEM) for exonuclease pausing and $0.58 \pm 0.02$ for replicative pausing, respectively.

### Reporting summary

Further information on research design is available in the Nature Portfolio Reporting Summary linked to this article.

## Data availability

The datasets generated and/or analyzed during the current study are available from Zenodo: https://doi.org/10.5281/zenodo.10782716[50]. Source data are provided with this paper.

## Code availability

The custom-written python scripts used in this study are available at https://github.com/longfuxu/DNAPolymeraseProject, under the MPL-2.0 license. The repository includes example dataset, example Jupiter notebook, along with a detailed README file for instructions on installation and usage.

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

## Acknowledgements

We thank Seyda Aca and Sandrine D'Haene for assistance with protein purification and DNA construction, Noémie Danné for help with implementing the step-fitting algorithm. We thank Mia Urem and Meindert Lamers for assistance with real-time DNA primer extension assay. We thank Erwin Peterman for critical reading and constructive feedback on this manuscript. This work was financially supported by a PhD fellowship from China Scholarship Council (To L.X., funding No. 201704910912), the European Union H2020 Marie-Sklowdowska Curie International Training Network AntiHelix (To G.J.L.W., funding No. 859853), and the European Research Council (ERC) under the European Union's Horizon 2020 research and innovation program MONOCHROME (to G.J.L.W., funding No.883240).

## Author contributions

L.X. and G.J.L.W. conceptualized the research. L.X. prepared protein samples and collected single-molecule data and analyzed data; M.T.J.H. analyzed data; L.X., M.T.J.H., and G.J.L.W. wrote and edited the manuscript; G.J.L.W. supervised the project; the manuscript is read, revised, and confirmed by all the listed authors.

## Competing interests

The combined optical tweezers and fluorescence technologies used in this article are patented and licensed to LUMICKS B.V., in which M.T.J.H. and G.J.L.W. declare a financial interest. The remaining authors declare no competing interests.
