## [Peer Review File · Nature Communications]

REVIEWER COMMENTS

Reviewer #1 (Remarks to the Author):

In this manuscript, Xu et al. report the dynamics of T7 polymerase during DNA replication using correlative optical tweezers fluorescence microscopy. They track the DNA extension changes as the polymerase performs DNA polymerization and cleavage, as well as binding and unbinding to the DNA. They show that the polymerase acts in a stochastic and auxiliary proteins-independent manner at the ssDNA/dsDNA junction, where it carries out exonucleolysis and polymerization. They also provide experimental evidence of the memory effect that contributes to the polymerase processivity. The authors employ a novel assay to investigate the polymerase dynamics and reveal the correlation between the force-dependent polymerase activity and the fluorescence signal. The manuscript is well-written and presents findings that are of broad interest for researchers in the field. The manuscript meets the standards of the journal and deserves publication after some revisions.

1. The authors should explain why exonuclease and polymerase activities are observed at high and low forces, respectively?
2. What is the rationale behind 60% labeling of polymerase?
3. Are DNA polymerase trajectories shown in the kymographs continuous? The overlay of the junction trajectory makes it difficult to visualize.
4. Are DNA polymerase always bound at the replication junction during the experiments?
5. Why do the authors use fairly high concentration of DNA polymerase, i.e., 30 nM for single-molecule fluorescence experiments? How does it impact the signal to noise ratio? Wouldn't the high polymerase concentration affect data interpretation while correlating the optical tweezers data and fluorescence signal?
6. Why the signal to noise ratio is low in the kymographs? Similar studies suggest a high signal to noise ratio can be achieved (Li, S., Wasserman, M.R., Yurieva, O. et al., Nat. Commun. 13, 2022; Lewis, Jacob S., et al., Mol. Cell 77, 2020).
7. What does the red box in Figure 1E represent?
8. In the kymographs (Figs. 1E, 3A, 4D and others), what does the high intensity at the bottom and top represent? In Figs. 4A and 4B, the authors indicate that the top and bottom intensity bands are attributed to signals from bead? Can authors clarify it and mention it in all the kymographs?
9. Figure 1, line 36: ... while the gray line represents fluorescence signal at the ssDNA/dsDNA junction. There is no gray line in Figure 1F?
10. What does the green trace at the back of the yellow trace in Fig. 1F represent?
11. The authors should mention the total length of the DNA handle in the main text.
12. Lines 202-204: This sentence should be used somewhere in the beginning to help authors understand the increasing/decreasing basepairs trends in Figure 1 and others.

Drs. Avinash Kumar and Yongli Zhang

Reviewer #2 (Remarks to the Author):

This work combines simultaneous confocal fluorescence and force extension techniques to track the activity of bacteriophage T7 DNA polymerase. Both exonuclease and polymerase activity are explored, where the T7 polymerase is labeled with a fluorophore. As in standard force spectroscopy experiments, changes in the fraction of single and double stranded DNA are tracked through the change in the overall length of the template as one is converted into the other. The new technique revealed in this work directly correlates length changes to the location of the T7 as imaged confocally and in kymographs. This enables a careful exploration of the rates of T7 stalling, release and even replacement on the template. While T7 polymerase activity has been widely characterized in force extension experiments, simultaneous imaging and force facilities answering questions that cannot be

answered by force spectroscopy alone, including how long each polymerase is bound and active.

Careful study of pauses and bursts of activity reveal that T7 binding and activity is surprisingly stochastic. The same polymerase may rapidly switch between active and inactive states, followed by release. The picture suggested by this data is not of a single, highly processive polymerase, but of a very rapid release and replacement by others from solution. Furthermore, many T7 appear to bind and never initiate polymerase/exonuclease activity. Together this indicates a 'memory' effect, where active polymerase/exonuclease activity is followed by more of the same and switching between the two modes is rarely seen. An intriguing explanation for this is proposed: that in solution, T7 exists in two conformations (and rarely switching), and unbinding T7 leaves the DNA junction in a specific state favorable only to T7 of the same conformation.

This is very interesting work that represents a significant advance over previous polymerase experiments and may suggest similar properties for polymerases from other systems. Furthermore, the data appears to be of high quality (though only a single fluorophore is tracked, and labeling is only 60% efficient), and is carefully analyzed. Finally, several controls for photobleaching and diffusion along single and double stranded DNA are shown. With minor clarifications, this paper should be suitable for publication.

Questions:

Figure 1 and Methods: Were the tagged T7 filtered after attachment?

Figure 1 and Methods: How large is the tag compared to the polymerase?

Figure 1 and Methods: What is the temporal resolution of the experiment? The pixel width is ~1 sec, but the methods and results mention that only events of 3 pixels were considered. So only pol exchanges that take longer than 3-5 sec can be seen? This should be discussed in detail.

Figure 1: The red box is not explained in the caption – is it for background subtraction?

Figure 2: There are a high percentage of non-fluorescent events that show exonuclease activity. Could any of this be due to DNA peeling at these high forces over these long times?

Figure 2: In panel E, the stated lifetimes in the caption do not match the values in the figure or in the text. In the caption, the values are distinct, but in the figure, the values are not distinct considering uncertainty.

For all figures, please check the panel ordering. It is confusing to follow panels that go in order right to left, then down, then back up to the top row.

The first equation is equation 3, but shouldn't this be started at 1?

Reviewer #3 (Remarks to the Author):

The authors present single molecule kinetic analysis of DNA polymerization by monitoring changes in length under tension on the template strand resulting from differences between single strand and double stranded DNA. By simultaneously monitoring fluorescence of labeled enzyme and polymerization, the authors describe exchange between polymerases during polymerization. The methods used by the authors are clever and the data appear to be interpreted rigorously. However, their method is unable to resolve single nucleotide addition steps and is limited to 10 ms per time

point. It is not the "high resolution" the authors claim. Moreover, the force on the DNA alters the kinetics as described below.

There is no description of the formation of a replication fork. The T7 DNA polymerase replication complex has been well characterized defining the coupled action of the helicase, polymerase, and exonuclease with leading and lagging strand synthesis (1). Rather it appears that the authors are only examining DNA elongation with a template and a primer with no duplex DNA and helicase ahead of the polymerase. The current study ignores these detailed studies and presents data that conflict with well-established properties of the polymerase.

The most severe limitation of the paper is the lack of quality controls for the changes induced by modification of the DNA polymerase. Most importantly, the authors use T7 gp5 polymerase fused with thioredoxin accessory protein. The only reference given in the text to the use of the fusion protein is a 1987 paper from Richardson's lab, which did not use a fusion protein. Studies have shown that without thioredoxin the DNA polymerase is no longer processive and "undergoes frequent dissociation from and rebinding to the DNA" (2), similar to the result reported in the current manuscript. No controls are given in this manuscript to assess the effect of fusing the thioredoxin to the polymerase. This is a fatal flaw since thioredoxin binding greatly affects polymerase activity and DNA dissociation rates. Fusing thioredoxin to achieve optimal interactions with the flexible DNA binding domain of the polymerase is not a trivial task and would require significant refinement and testing by established methods to define polymerization rates and processivity. A likely explanation of the results presented in this paper is that a less than optimal covalent linkage of thioredoxin to the polymerase was insufficient to stabilize an otherwise flexible DNA-binding domain leading thereby failing to achieve the desired tight DNA binding.

The expression of the T7 pol-thioredoxin fusion protein with an additional SNAP tag (182 amino acids, 19.4 kDa) is also not well described. Was the SNAP tag added to the N- or C- terminus of T7 gp5 and was a flexible linker included? The SNAP tag is likely to introduce a significant change to the polymerase structure and dynamics for which there are no controls. The addition of a large fluorophore further compounds problems. T7 DNA polymerase is a dynamic enzyme where conformational changes dictate nucleotide specificity and proofreading. T7 DNA polymerase has been extensively characterized by single turnover kinetic methods with single base-pair resolution, far exceeding the capabilities of single molecule methods to define reaction kinetics and mechanism (3-9). The results of numerous studies are at odds with the authors' conclusions. The enzyme catalyzes replication at ~300 bp/s and dissociates from DNA at with a rate constant of 0.2/s giving a processivity of 1500 bp. The exonuclease reaction on ssDNA occurs at 1000/s but with duplex DNA the exonuclease is governed by the rate of transfer of the primer strand from the polymerase to the exonuclease site. The kinetic partitioning between polymerization and exonuclease reaction has also been well defined. These studies should stand as a benchmark for evaluating any labeled enzyme to quantify the effects of modifications due to fusing with other proteins and adding fluorescent labels.

For each of these issues, the authors must address the effects of their significant enzyme modifications. Otherwise, their results only apply to the modified enzyme and are not applicable to polymerase function in the cell. At the very minimum, the following experimental measurements are necessary. Single turnover kinetic studies in solution can be easily performed with native and modified enzymes to quantify the effects of modifications.

- a. What is the active site concentration of the modified, labeled enzyme? That is, what fraction of the enzyme still binds and extends DNA?
- b. What is the rate of polymerization in solution?
- c. What is the K_d for DNA binding to the altered enzyme?
- d. Most importantly, what is the rate constant for DNA dissociation?

The effects of force on the DNA may be responsible for some of the unusual behavior seen in this report. Structural studies show that the template DNA enters the active site at 90 degrees to the

growing duplex (10). Therefore, as suggested in the present studies and many previous reports using this method, adding tension to the two ends of the DNA distorts the polymerase active site. It is remarkable that the authors show that a force of 40–50 pN disrupts the polymerase activity so that only exonuclease reaction is seen. But they assume that dropping the force only 2–4-fold to 10–20 pN allows measurement of polymerization without affecting fundamental kinetic parameters including the polymerization and DNA dissociation rate. The authors' analysis rests on an unsupported assumption, and it is likely that the unique conclusions put forth by the authors regarding fast dissociation of the polymerase are an artefact of their methods and enzyme preparation, and there are no controls offered to overcome this objection.

Memory effects have been proposed from single molecule kinetic studies, but to my knowledge there has never been definitive data to support such postulates. Enzymes do not have memories; they are governed by rate constants for transitioning between states.

Literature Cited

1. Singh, A., and Patel, S. S. (2022) Quantitative methods to study helicase, DNA polymerase, and exonuclease coupling during DNA replication *Methods Enzymol* 672, 75-102
10.1016/bs.mie.2022.03.011
2. Etsen, C. M., Hamdan, S. M., Richardson, C. C., and van Oijen, A. M. (2010) Thioredoxin suppresses microscopic hopping of T7 DNA polymerase on duplex DNA *Proc Natl Acad Sci U S A* 107, 1900-1905
10.1073/pnas.0912664107
3. Dangerfield, T. L., and Johnson, K. A. (2021) Conformational dynamics during high-fidelity DNA replication and translocation defined using a DNA polymerase with a fluorescent artificial amino acid *Journal of Biological Chemistry* 296, 100143-100161 10.1074/jbc.RA120.016617
4. Dangerfield, T. L., Kirmizialtin, S., and Johnson, K. A. (2022) Conformational dynamics during misincorporation and mismatch extension defined using a DNA polymerase with a fluorescent artificial amino acid *J Biol Chem* 298, 101451 10.1016/j.jbc.2021.101451
5. Dangerfield, T. L., Kirmizialtin, S., and Johnson, K. A. (2022) Substrate specificity and proposed structure of the proofreading complex of T7 DNA polymerase *J Biol Chem* 298, 101627
10.1016/j.jbc.2022.101627
6. Dangerfield, T. L., and Johnson, K. A. (2023) Kinetics of DNA strand transfer between polymerase and proofreading exonuclease active sites regulates error correction during high-fidelity replication *Journal of Biological Chemistry* 299, 102744 10.1016/j.jbc.2022.102744
7. Donlin, M. J., Patel, S. S., and Johnson, K. A. (1991) Kinetic partitioning between the exonuclease and polymerase sites in DNA error correction *Biochemistry* 30, 538-546, PM:1988042
8. Patel, S. S., Wong, I., and Johnson, K. A. (1991) Pre-Steady-State Kinetic-Analysis of Processive DNA-Replication Including Complete Characterization of An Exonuclease-Deficient Mutant *Biochemistry* 30, 511-525, ISI:A1991ET38100029
9. Wong, I., Patel, S. S., and Johnson, K. A. (1991) An induced-fit kinetic mechanism for DNA replication fidelity: direct measurement by single-turnover kinetics *Biochemistry* 30, 526-537, PM:1846299
10. Doublie, S., Tabor, S., Long, A. M., Richardson, C. C., and Ellenberger, T. (1998) Crystal structure of a bacteriophage T7 DNA replication complex at 2.2 Å resolution *Nature* 391, 251-258, PM:9440688

Response to Reviewers' Comments

Reviewer #1 (Remarks to the Author):

In this manuscript, Xu et al. report the dynamics of T7 polymerase during DNA replication using correlative optical tweezers fluorescence microscopy. They track the DNA extension changes as the polymerase performs DNA polymerization and cleavage, as well as binding and unbinding to the DNA. They show that the polymerase acts in a stochastic and auxiliary proteins-independent manner at the ssDNA/dsDNA junction, where it carries out exonucleolysis and polymerization. They also provide experimental evidence of the memory effect that contributes to the polymerase processivity. The authors employ a novel assay to investigate the polymerase dynamics and reveal the correlation between the force-dependent polymerase activity and the fluorescence signal. The manuscript is well-written and presents findings that are of broad interest for researchers in the field. The manuscript meets the standards of the journal and deserves publication after some revisions.

1. The authors should explain why exonuclease and polymerase activities are observed at high and low forces, respectively?

We appreciate the reviewer's query regarding the force-dependent activities of DNA polymerase. The observed phenomena stem from the mechanical forces affecting the structure of the nucleic acids. At higher forces (i.e.: ~50 pN), the DNA molecule is stretched, which can expose the ends of the DNA strands, making them more accessible to exonucleases, removing nucleotides from the ends of DNA strands. This is crucial for removing damaged or mismatched nucleotides. Conversely, at lower forces (i.e.: ~ 20 pN), the DNA structure is more relaxed, allowing polymerases to effectively match and add nucleotides to the growing strand. Low mechanical stress ensures accurate base pairing and efficient strand elongation. This force-dependent behavior has been reported regularly over the last two decades, see for example these earlier studies on T7 DNA polymerase¹⁻³. We now give brief background and suitable references about this in **Methods** section, with changes marked in red.

2. What is the rationale behind 60% labeling of polymerase?

The choice of a 60% labeling ratio for the polymerase was strategic to allow effective visualization of protein exchange at the replication fork. This incomplete labeling approach ensures that not every polymerase molecule is fluorescent, enabling the detection of new, unlabeled polymerases replacing labeled ones, thereby providing direct evidence of polymerase exchange dynamics.

3. Are DNA polymerase trajectories shown in the kymographs continuous? The overlay of the junction trajectory makes it difficult to visualize.

We acknowledge the reviewer's concern about the clarity of kymographs. The trajectories of DNA polymerase in the kymographs are not continuous due to the dynamic nature of polymerase binding and unbinding and the incomplete labelling. We now added the raw kymograph without the overlay as a panel in **Figure S2A** for clarity.

4. Are DNA polymerase always bound at the replication junction during the experiments?

During our experiments, DNA polymerase is not continuously bound at the replication junction. Our findings demonstrate that the polymerase undergoes rapid binding and unbinding events, contributing to the dynamic and stochastic nature of the replication

process. This can also be observed by quantifying the fluorescence intensity along the ssDNA/dsDNA junction (**Figure 1F** and **1G**).

In addition to binding at the ssDNA/dsDNA junction, DNA polymerases were also observed to bind on dsDNA, potentially searching for binding sites (**Figure 4A**). Finally, DNA polymerases were observed to bind stationary on ssDNA (**Figure 4B**).

5. Why do the authors use fairly high concentration of DNA polymerase, i.e., 30 nM for single-molecule fluorescence experiments? How does it impact the signal to noise ratio? Wouldn't the high polymerase concentration affect data interpretation while correlating the optical tweezers data and fluorescence signal?

The 30 nM concentration of DNA polymerase was optimized and chosen to ensure a sufficient frequency of binding events and continuous activity for robust statistical analysis, following earlier studies^{1,3}. In our experiments, a lower concentration of DNA polymerase could lead to reduced apparent processivity, potentially making it more challenging to detect these events. And importantly, we use confocal microscopy for imaging the DNA which provides a good signal to noise ratio because it images the fluorescent proteins that are on the DNA template while rejecting most of the background.

6. Why the signal to noise ratio is low in the kymographs? Similar studies suggest a high signal to noise ratio can be achieved (Li, S., Wasserman, M.R., Yurieva, O. et al., Nat. Commun. 13, 2022; Lewis, Jacob S., et al., Mol. Cell 77, 2020).

We appreciate the reviewers for their query regarding the signal-to-noise ratio (SNR) in our kymographs. It's important to note that the mentioned two publications investigate DNA polymerase systems that appear to have long lifetimes on the DNA; moreover, they were able to use different fluorophores with distinct imaging parameters. The SNR in our experiments is influenced by several factors, including the fluorescence intensity of labeled proteins (dye quantum yield), and the dynamic nature of polymerase interactions with DNA. In our experiments, we optimized labeling to achieve a balance between detectable signal and maintaining the native functionality of the polymerase, as well as to observe the dynamics of DNA polymerase. In our earlier trials, over-labeling or using bright fluorophores (such as Atto647N) appeared to alter the protein dynamics. In addition, we think that the rapid and transient interactions of DNA polymerase with the DNA substrate can lead to fluctuating signal intensities. We now clarify this point more explicitly in the Method section.

7. What does the red box in Figure 1E represent?

The red box in Figure 1E is utilized for background subtraction in our analysis. It defines a specific region in the kymograph where the background fluorescence signal is measured. This background signal is then subtracted from the fluorescence signal of the DNA polymerase to enhance the accuracy of our signal-to-noise ratio and ensure that the fluorescence changes we observe are indeed due to the binding and activity of the DNA polymerase, not background fluorescence. We clarified this in the figure caption.

8. In the kymographs (Figs. 1E, 3A, 4D and others), what does the high intensity at the bottom and top represent? In Figs. 4A and 4B, the authors indicate that the top and bottom intensity bands are attributed to signals from bead? Can authors clarify it and mention it in all the kymographs?

The high-intensity bands at the top and bottom of the kymographs are indeed due to fluorescence signals from the beads used in our optical tweezer's setup. These signals are not related to the DNA polymerase activity and serve as reference points in the kymographs.

This information is now clearly mentioned in all relevant figure captions for better understanding in the revised manuscript.

9. Figure 1, line 36: ... while the gray line represents fluorescence signal at the ssDNA/dsDNA junction. There is no gray line in Figure 1F?

We apologize for the confusion. The mentioned gray line is the raw data of fluorescence signal of DNA polymerase at the ssDNA/dsDNA junction. To ensure the clarity, we revised the caption to "*while the yellow line represents the filtered fluorescence signal of DNA polymerase at the ssDNA/dsDNA junction (raw data in gray)*"

10. What does the green trace at the back of the yellow trace in Fig. 1F represent?

As explained in comments #9, the gray trace (instead of green trace) at the back of the yellow line represents the raw fluorescence data, directly extracted from the DNA polymerase trajectory. In contrast, the yellow line illustrates the filtered signal using a Savitzky-Golay filter, as detailed in our methods section. This approach was employed to enhance the clarity and interpretability of the fluorescence signal, facilitating a more precise correlation with the DNAP's mechanical activities. We revised the caption for clarity.

11. The authors should mention the total length of the DNA handle in the main text.

The total length of the DNA handle used in our experiments was 8393 base pairs. We now included this information in the main text of the revised manuscript.

12. Lines 202-204: This sentence should be used somewhere in the beginning to help authors understand the increasing/decreasing basepairs trends in Figure 1 and others.

We agree that providing this explanation earlier in the text will aid readers in understanding the context of our findings. We now incorporate this sentence at an appropriate earlier section in the revised manuscript.

Drs. Avinash Kumar and Yongli Zhang

We thank both the reviewers for their critical and constructive comments.

Reviewer #2 (Remarks to the Author):

This work combines simultaneous confocal fluorescence and force extension techniques to track the activity of bacteriophage T7 DNA polymerase. Both exonuclease and polymerase activity are explored, where the T7 polymerase is labeled with a fluorophore. As in standard force spectroscopy experiments, changes in the fraction of single and double stranded DNA are tracked through the change in the overall length of the template as one is converted into the other. The new technique revealed in this work directly correlates length changes to the location of the T7 as imaged confocally and in kymographs. This enables a careful exploration of the rates of T7 stalling, release and even replacement on the template. While T7 polymerase activity has been widely characterized in force extension experiments, simultaneous imaging and force facilities answering questions that cannot be answered by force spectroscopy alone, including how long each polymerase is bound and active.

Careful study of pauses and bursts of activity reveal that T7 binding and activity is surprisingly stochastic. The same polymerase may rapidly switch between active and inactive states, followed by release. The picture suggested by this data is not of a single, highly processive polymerase, but of a very rapid release and replacement by others from solution. Furthermore, many T7 appear to bind and never initiate polymerase/exonuclease activity. Together this indicates a 'memory' effect, where active polymerase/exonuclease activity is followed by more of the same and switching between the two modes is rarely seen. An intriguing explanation for this is proposed: that in solution, T7 exists in two conformations (and rarely switching), and unbinding T7 leaves the DNA junction in a specific state favorable only to T7 of the same conformation.

This is very interesting work that represents a significant advance over previous polymerase experiments and may suggest similar properties for polymerases from other systems. Furthermore, the data appears to be of high quality (though only a single fluorophore is tracked, and labeling is only 60% efficient), and is carefully analyzed. Finally, several controls for photobleaching and diffusion along single and double stranded DNA are shown. With minor clarifications, this paper should be suitable for publication.

Questions:

1. Figure 1 and Methods: Were the tagged T7 filtered after attachment?

We appreciate the reviewer's query regarding the filtration of tagged T7 DNA polymerase. Following the attachment of the fluorophore to the T7 DNA polymerase, the labelled polymerase was filtered. This step ensures the removal of any unbound or excess fluorophores, thereby reducing background fluorescence and enhancing the specificity of our imaging. The details of this process are revised in the 'Fluorescently labelled DNA Polymerase Preparation and DNA Template Construction' section of the Methods.

2. Figure 1 and Methods: How large is the tag compared to the polymerase?

The tag used in our study is relatively small compared to that of the T7 polymerase. Specifically, the SNAP-tag, which is ~ 20 KDa in size, is fused to the T7 DNA polymerase (approximate molecular weight of ~96 KDa). We now added this value in the method section for clarity.

3. Figure 1 and Methods: What is the temporal resolution of the experiment? The pixel width is ~1 sec, but the methods and results mention that only events of 3 pixels were considered. So only pol exchanges that take longer than 3-5 sec can be seen? This should be discussed in detail.

There might be a bit of unclarity about the temporal resolution of the instrument. The temporal resolution of our image data was determined by the scanning rate of confocal microscopy which results in a pixel width that is approximately 0.3 sec. In our analysis, only the events lasting >3 pixels (~1 sec) were considered for robust analysis, to minimize artifact from diffusive DNA polymerase in solution and ensure the capture of significant binding events. The temporal resolution, therefore, represents a deliberate choice to ensure robust and meaningful data analysis and is around 1 sec instead of the assumed 3-5 sec by the reviewer. We rewrote the text to enhance clarity about this point.

4. Figure 1: The red box is not explained in the caption – is it for background subtraction?

We apologize for the confusion. The red box in Figure 1 is indeed used for background fluorescence subtraction. We now clarified this in the figure caption.

5. Figure 2: There are a high percentage of non-fluorescent events that show exonuclease activity. Could any of this be due to DNA peeling at these high forces over these long times?

This is a great point, and we appreciate the reviewer's query in this regard. However, several aspects of our experimental design and observations suggest that the observed events are due to genuine DNA polymerase activities rather than DNA peeling.

Firstly, our experiments were conducted under controlled tension conditions, specifically optimized to observe the activities of T7 DNA polymerase. The forces applied (45-50 pN for exonuclease activity and 10-20 pN for polymerase activity) were within the range to promote distinct polymerase and exonuclease activities without inducing DNA peeling. DNA peeling, while a potential occurrence under high tension (> 65 pN) with low ionic strength conditions^{4,5}, typically results in abrupt and significant changes in the DNA length, which were not observed in our data. In our experimental design, the method of DNA tethering and buffer conditions with 100 mM NaCl and 3 mM MgCl₂, minimizes the likelihood of DNA peeling.

Further, our analysis of the binding lifetimes of DNA polymerase (Figure 2E) indicates that the observed exonuclease activities are consistent with the behavior of T7 DNA polymerase. The binding lifetimes are in line with what is expected for DNA polymerase functioning rather than DNA peeling.

Last, the non-fluorescent events showing exonuclease activity were consistently correlated with the behavior observed in fluorescent events. This correlation strengthens the argument that these events are representative of bona fide DNA polymerase activities.

While the possibility of DNA peeling is a valid concern, the evidence from our controlled experimental setup and the correlation of non-fluorescent events with fluorescent ones strongly supports that the observed exonuclease activities are due to DNA polymerase action. We have now added the following sentences to the revised manuscript to explain this better.

"Notably, the applied forces (45-50 pN for exonuclease and 10-20 pN for polymerase activity), combined with our DNA tethering method and buffer conditions (100 mM NaCl and 3 mM MgCl₂), strongly supports that the observed exonuclease activities are due to DNA polymerase action, instead of DNA peeling, which typically occurs under high tension (>65 pN) with low ionic strength conditions. ^{4,5}"

6. Figure 2: In panel E, the stated lifetimes in the caption do not match the values in the figure or in the text. In the caption, the values are distinct, but in the figure, the values are not distinct considering uncertainty.

We thank the reviewer for the careful examination. In the caption, we stated, "*The average lifetime for DNA polymerase under exonuclease events is measured to be $t = 1.34 \pm 0.06$ s, and under polymerase events, it is $t = 1.11 \pm 0.08$ s.*" However, in the figure, we attempted for a representation with one decimal place for clarity and conciseness, thus rounding off these values to 1.3 s and 1.1 s, respectively. The earlier choice was to maintain the figure's readability and avoid overloading it with excessive numerical precision that might not significantly impact the interpretation of the data. We have now reviewed the representations across the figure, caption, and main text to ensure consistency and clarity. The revised values accurately reflect the data and are consistent across all elements of the manuscript.

7. For all figures, please check the panel ordering. It is confusing to follow panels that go in order right to left, then down, then back up to the top row.

We acknowledge that the current ordering of panels might be confusing, and we appreciate your suggestion to improve readability. In the revised manuscript, we revised the panel ordering of these figures.

8. The first equation is equation 3, but shouldn't this be started at 1?

We appreciate your attention to the equation numbering in our manuscript. The numbering of equations starts from the Methods section, where two equations precede the one referenced as Equation (3) in the main text. These two earlier equations (1) and (2) are critical for establishing the foundational calculations that lead to Equation (3) and thus were referred earlier in the main text. This numbering was chosen to maintain a logical and sequential flow of the mathematical derivations presented.

Reviewer #3 (Remarks to the Author):

The authors present single molecule kinetic analysis of DNA polymerization by monitoring changes in length under tension on the template strand resulting from differences between single strand and double stranded DNA. By simultaneously monitoring fluorescence of labeled enzyme and polymerization, the authors describe exchange between polymerases during polymerization. The methods used by the authors are clever and the data appear to be interpreted rigorously.

(1) However, their method is unable to resolve single nucleotide addition steps and is limited to 10 ms per time point. It is not the “high resolution” the authors claim. Moreover, the force on the DNA alters the kinetics as described below.

We agree that a relative term such as “high resolution” is not useful and might even distract from the actual strength of the method. Hence, as suggested, we removed the term “high resolution”.

Regarding your concern about the applied force on the DNA template affecting kinetics, we address this point in detail in our response to Comment #4.

(2) There is no description of the formation of a replication fork. The T7 DNA polymerase replication complex has been well characterized defining the coupled action of the helicase, polymerase, and exonuclease with leading and lagging strand synthesis (1). Rather it appears that the authors are only examining DNA elongation with a template and a primer with no duplex DNA and helicase ahead of the polymerase. The current study ignores these detailed studies and presents data that conflict with well-established properties of the polymerase.

We thank the reviewer for this comment. In our approach, we deliberately employed a simplified system to isolate and closely examine the intrinsic exchange dynamics of DNA polymerase independent of helicase action and other replication machinery. This methodological choice was made to specifically explore whether DNA polymerase can exchange autonomously, which is challenging to study in more complex replication systems. Our study does not contradict the established roles and interactions of polymerase, helicase, and exonuclease but rather complements these findings by offering insights into the polymerase’s behavior in a controlled environment.

We recognize that this simplification might raise questions about the relevance of our findings to the more complex in vivo replication machinery. To address this, we revised the manuscript to include a clearer explanation of our experimental approach and its rationale in Introduction and Methods. In the Discussion, we also discussed how our findings on the autonomous exchange dynamics of DNA polymerase relate to the established knowledge of T7 replication fork dynamics in the context of error correction by DNA polymerase. We hope this will provide readers with a comprehensive view of the polymerase’s behavior both as an independent entity and as part of the larger replication complex.

(3) The most severe limitation of the paper is the lack of quality controls for the changes induced by modification of the DNA polymerase. Most importantly, the authors use T7 gp5 polymerase fused with thioredoxin accessory protein. The only reference given in the text to the use of the fusion protein is a 1987 paper from Richardson’s lab, which did not use a fusion protein. Studies have shown that without thioredoxin the DNA polymerase is no longer processive and “undergoes frequent dissociation from and rebinding to the DNA” (2), similar to the result reported in the current manuscript. No controls are given in this manuscript to assess the effect of fusing the thioredoxin to the polymerase. This is a fatal flaw since

thioredoxin binding greatly affects polymerase activity and DNA dissociation rates. Fusing thioredoxin to achieve optimal interactions with the flexible DNA binding domain of the polymerase is not a trivial task and would require significant refinement and testing by established methods to define polymerization rates and processivity. A likely explanation of the results presented in this paper is that a less than optimal covalent linkage of thioredoxin to the polymerase was insufficient to stabilize an otherwise flexible DNA-binding domain leading thereby failing to achieve the desired tight DNA binding.

The expression of the T7 pol-thioredoxin fusion protein with an additional SNAP tag (182 amino acids, 19.4 kDa) is also not well described. Was the SNAP tag added to the N- or C-terminus of T7 gp5 and was a flexible linker included? The SNAP tag is likely to introduce a significant change to the polymerase structure and dynamics for which there are no controls. The addition of a large fluorophore further compounds problems. T7 DNA polymerase is a dynamic enzyme where conformational changes dictate nucleotide specificity and proofreading. T7 DNA polymerase has been extensively characterized by single turnover kinetic methods with single base-pair resolution, far exceeding the capabilities of single molecule methods to define reaction kinetics and mechanism (3-9). The results of numerous studies are at odds with the authors' conclusions. The enzyme catalyzes replication at ~300 bp/s and dissociates from DNA at with a rate constant of 0.2/s giving a processivity of 1500 bp. The exonuclease reaction on ssDNA occurs at 1000/s but with duplex DNA the exonuclease is governed by the rate of transfer of the primer strand from the polymerase to the exonuclease site. The kinetic partitioning between polymerization and exonuclease reaction has also been well defined. These studies should stand as a benchmark for evaluating any labeled enzyme to quantify the effects of modifications due to fusing with other proteins and adding fluorescent labels.

For each of these issues, the authors must address the effects of their significant enzyme modifications. Otherwise, their results only apply to the modified enzyme and are not applicable to polymerase function in the cell. At the very minimum, the following experimental measurements are necessary. Single turnover kinetic studies in solution can be easily performed with native and modified enzymes to quantify the effects of modifications.

- a. What is the active site concentration of the modified, labeled enzyme? That is, what fraction of the enzyme still binds and extends DNA?
- b. What is the rate of polymerization in solution?
- c. What is the K_d for DNA binding to the altered enzyme?
- d. Most importantly, what is the rate constant for DNA dissociation?

We appreciate the reviewer's insightful comments on the potential impact of the modifications made to the T7 DNA polymerase in our research. Our study employs a modified T7 DNA polymerase, which includes a SNAP tag for fluorescence labeling and a fusion to thioredoxin. These modifications are crucial for single-molecule analysis and were meticulously carried out to preserve the enzyme's functionality. Comparative analyses with other studies, both single-molecule and ensemble approaches, corroborate the validity and significance of our results. See the details about this below.

The fluorescence labelling of T7 DNA polymerase was necessary for our single-molecule analysis. Direct labeling of the polymerase at the C-terminal, N-terminal, or cysteine residues often resulted in inactivity in our trials. Consequently, we opted for an N-terminal SNAP-tag. Although this tag slightly increases the polymerase's molecular weight (by ~20%), the use of SNAP-tag fusion proteins for fluorescence labeling is a widely recognized technique in bioimaging both in vivo and in vitro studies^{6,7}. The choice of the labeling site and fluorophore size were carefully considered to minimize interference with the enzyme's active site and conformational changes (see **Methods**). We also compare the properties of our modified

DNA polymerase with that of the commercial polymerase (**Figure R1** and **Table R1**) showing similar results.

The fusion of *trx* to DNA polymerase aims to preserve DNA polymerase activity after SNAP-tag fusion and to avoid any potential inactivity of polymerase. In our study, we cloned gp5 from T7 and *trxA* gene from *E. coli* into the pET-Duet1 multiple cloning sites. A flexible linker was included to minimize steric hindrance and maintain enzyme functionality. Our modifications to established protocols^{8,9} were designed to maintain the native characteristics of the polymerase while meeting the specific requirements of our experimental setup. The reference to the 1987 Richardson lab paper⁹ was to acknowledge the origin of this approach. We intended to also cite this publication from 2003 Richardson lab paper⁸, which demonstrated a covalently linked DNA polymerase with *trx* showing comparable activity between covalently linked and infusion versions of gp5+*trx*. We revised our Method section to clarify this.

To ensure the recombined DNA polymerase functioned as close to its native state as possible, we added 625 nM dNTPs (mix) and 2-4-fold excess *trx*, as supported by previous publications from Richardson lab and Johnson Lab^{8,10}. Through these steps, we ensure the fused DNA polymerase retain its native processivity. This information, initially in the raw data metadata, is now detailed in the revised manuscript's Methods section.

Lacking access to radio-labelled nucleotides and detection methods for single turnover assays, we employed single-molecule methods for activity test and comparison, yielding results comparable to published data using the same methods in the same lab. Thus our T7 fused DNA polymerase-thioredoxin (**Figure R1 C&D**) displays kinetics that is more or less the same as the commercially available DNA polymerase of the published data (see **Figure R1 A&B**)³.

Figure R1. Comparative Analysis of Enzymatic Activity Between Commercial T7 DNA Polymerase (Control) (panels A and B), adapted from ref ³ and Modified DNA Polymerase (Experimental) (panels C and D). (A) and (C): These panels illustrate the polymerization and exonucleolysis activities of the commercial and modified DNA polymerases, respectively. (B) and (D): These panels depict the relative probability of DNA polymerase binding with its exonuclease (exo) active site for both the commercial and modified enzymes.

Moreover, as suggested by the reviewer, we compared our single-molecule data with ensemble research and other single-molecule studies to ensure consistency and reliability (see **Table R1**). Our replication rate, while not identical to previous findings, aligns well within expected ranges. Direct observation of on-rate and off-rate provided a unique assessment of processivity. The slight differences in these values may be attributed to variables such as template length, applied tension, and solution conditions (e.g., ion strength and crowding agents).

	Concentration of DNAP (nM)	Replication Rate (bp/s)	Dissociation Rate (-s)	Example Refs
Ensemble Assay	1000 nM	~ 300 bp/s without tension	0.2 s ⁻¹	10,11
Single-molecule studies from literatures	0.8–880 nM	100-400 bp/s with tension	0.6-1.3 s ⁻¹	1,12
Current study	30nM	100~350 bp/s with tension	0.75-0.9 s ⁻¹	Current report

As a final note, in this report, our primary objective was to correlate kinetic activity (measured through changes in DNA template length) with real-time fluorescence monitoring. This method, demonstrated in our report, can be beneficial for many other single-molecule studies investigating DNA-protein interactions. The conclusions drawn in fact align well with recent findings on DNA replication in T7 replisomes^{13–15}, bacteria^{16–22}, and eukaryotes^{23,24}.

(4) The effects of force on the DNA may be responsible for some of the unusual behavior seen in this report. Structural studies show that the template DNA enters the active site at 90 degrees to the growing duplex (10). Therefore, as suggested in the present studies and many previous reports using this method, adding tension to the two ends of the DNA distorts the polymerase active site. It is remarkable that the authors show that a force of 40–50 pN disrupts the polymerase activity so that only exonuclease reaction is seen. But they assume that dropping the force only 2–4-fold to 10–20 pN allows measurement of polymerization without affecting fundamental kinetic parameters including the polymerization and DNA dissociation rate. The authors' analysis rests on an unsupported assumption, and it is likely that the unique conclusions put forth by the authors regarding fast dissociation of the polymerase are an artefact of their methods and enzyme preparation, and there are no controls offered to overcome this objection.

We appreciate reviewer's remarks regarding the potential force effects on the DNA substrate and its implications for our observations. Firstly, while our study applies an external force to the DNA substrate, it's important to note that in cellular environments, DNA template is also influenced by tension^{25,26}. This tension can be induced by protein-bound events or structural changes within the DNA. For instance, when a mismatch is incorporated into the DNA duplex, it can induce tension/stress at the replication junction^{27,28}. Such tension could

facilitate the transition of DNA polymerase to function in an exonuclease mode for proofreading²⁹. Such forces on DNA template are challenging to study with conventional methods. Our approach, therefore, although simplified, attempts to mirror certain aspects of the dynamic mechanical environment that DNA polymerase encounters *in vivo*.

Regarding your concern about the predominance of exonuclease activity at higher forces (~50 pN) and polymerization activity at reduced forces (10-20 pN). This effect is already known and reported for a long time. For instance, Wuite et al. demonstrated similar behavior in T7 DNA polymerase under mechanical forces¹. Worth mentioning, these force-dependent activity is not specific to T7 DNA polymerase, studies on T4 DNA polymerase³⁰, *E. coli* DNA polymerase³¹, mitochondrial DNA polymerase³², and Φ 29 DNA polymerase²⁹ show similar results. Our measurements of DNA polymerase binding duration at higher (i.e.:50 pN) and lower (i.e.: 20 pN) tension revealed comparable durations and dissociation rates with earlier studies (**Figure 2E**). Moreover, it is essential to note that while the applied force is an external factor, it does not necessarily distort the enzyme's active site directly. The force primarily affects the DNA template's conformation, which in turn could influence enzyme activity indirectly. The ranges of forces we used were chosen based on established literature^{1,12,29-32}.

We acknowledge that the kinetics observed in our study, under the influence of applied force, may differ from the enzyme's behavior in a single turnover assay and we recognized that both methods are not the same as the natural cellular environment. However, our observations provide valuable insights into how tension, whether externally applied or intrinsically generated, can influence DNA polymerase dynamics. This aspect of our study adds to the understanding of DNA polymerase functionality under varying mechanical conditions, which is pertinent to both *in vitro* and *in vivo* scenarios. Plus, the potential differences in kinetics observed in our study compared to ensemble biochemistry studies without tension offer valuable information about the enzyme's adaptability and mechanosensitivity.

We presented these points clearer in our revised manuscript to ensure that the implications of our findings are appropriately contextualized within the broader framework of DNA replication dynamics, particularly emphasizing the role of tension in regulating DNA polymerase activity.

(5) Memory effects have been proposed from single molecule kinetic studies, but to my knowledge there has never been definitive data to support such postulates. Enzymes do not have memories; they are governed by rate constants for transitioning between states.

We recognize the reviewers' concerns about the "memory effect. Indeed, enzymes, being non-sentient entities, do not have memories in the conventional sense. However, what we describe as a 'memory effect' is a metaphorical way of interpreting the observed phenomena where the behavior of DNA polymerase upon rebinding at the replication fork appears to maintain its previous state (either engaged in polymerization, exonuclease activity, or a paused state). Our data revealed that the catalytic activity of DNA polymerase is preserved over a longer time than the actual binding time of a single protein molecule, leading to the chemically continuous yet kinetically discontinuous nature of replication. We now further clarify this in the revised manuscript.

Literature Cited

1. Singh, A., and Patel, S. S. (2022) Quantitative methods to study helicase, DNA polymerase, and exonuclease coupling during DNA replication *Methods Enzymol* 672, 75-102
10.1016/bs.mie.2022.03.011
2. Etson, C. M., Hamdan, S. M., Richardson, C. C., and van Oijen, A. M. (2010) Thioredoxin suppresses

- microscopic hopping of T7 DNA polymerase on duplex DNA Proc Natl Acad Sci U S A 107, 1900-1905
10.1073/pnas.0912664107
3. Dangerfield, T. L., and Johnson, K. A. (2021) Conformational dynamics during high-fidelity DNA replication and translocation defined using a DNA polymerase with a fluorescent artificial amino acid Journal of Biological Chemistry 296, 100143-100161 10.1074/jbc.RA120.016617
 4. Dangerfield, T. L., Kirmizialtin, S., and Johnson, K. A. (2022) Conformational dynamics during misincorporation and mismatch extension defined using a DNA polymerase with a fluorescent artificial amino acid J Biol Chem 298, 101451 10.1016/j.jbc.2021.101451
 5. Dangerfield, T. L., Kirmizialtin, S., and Johnson, K. A. (2022) Substrate specificity and proposed structure of the proofreading complex of T7 DNA polymerase J Biol Chem 298, 101627 10.1016/j.jbc.2022.101627
 6. Dangerfield, T. L., and Johnson, K. A. (2023) Kinetics of DNA strand transfer between polymerase and proofreading exonuclease active sites regulates error correction during high-fidelity replication Journal of Biological Chemistry 299, 102744 10.1016/j.jbc.2022.102744
 7. Donlin, M. J., Patel, S. S., and Johnson, K. A. (1991) Kinetic partitioning between the exonuclease and polymerase sites in DNA error correction Biochemistry 30, 538-546, PM:1988042
 8. Patel, S. S., Wong, I., and Johnson, K. A. (1991) Pre-Steady-State Kinetic-Analysis of Processive DNA-Replication Including Complete Characterization of An Exonuclease-Deficient Mutant Biochemistry 30, 511-525, ISI:A1991ET38100029
 9. Wong, I., Patel, S. S., and Johnson, K. A. (1991) An induced-fit kinetic mechanism for DNA replication fidelity: direct measurement by single-turnover kinetics Biochemistry 30, 526-537, PM:1846299
 10. Double, S., Tabor, S., Long, A. M., Richardson, C. C., and Ellenberger, T. (1998) Crystal structure of a bacteriophage T7 DNA replication complex at 2.2 Å resolution Nature 391, 251-258, PM:9440688

We thank the reviewer for providing these literatures. We have carefully examined and compared their studies with our results and have include the relevant literatures in our revised manuscript and responses.

References

1. Wuite, G. J. L., Smith, S. B., Young, M., Keller, D. & Bustamante, C. Single-molecule studies of the effect of template tension on T7 DNA polymerase activity. *Nature* **404**, 103–106 (2000).
2. Hamdan, S. M. *et al.* Dynamic DNA helicase-DNA polymerase interactions assure processive replication fork movement. *Mol Cell* **27**, 539–549 (2007).
3. Hoekstra, T. P. *et al.* Switching between Exonucleolysis and Replication by T7 DNA Polymerase Ensures High Fidelity. *Biophys J* **112**, 575–583 (2017).
4. Gross, P. *et al.* Quantifying how DNA stretches, melts and changes twist under tension. *Nature Physics* **7**, 731–736 (2011).
5. Broekmans, O. D., King, G. A., Stephens, G. J. & Wuite, G. J. L. DNA Twist Stability Changes with Magnesium(2+) Concentration. *Physical Review Letters* **116**, (2016).
6. Cole, N. B. Site-Specific Protein Labeling with SNAP-Tags. *Curr Protoc Protein Sci* **73**, 30.1.1-30.1.16 (2013).
7. Corrêa, I. R. *et al.* Substrates for improved live-cell fluorescence labeling of SNAP-tag. *Curr Pharm Des* **19**, 5414–5420 (2013).
8. Johnson, D. E. & Richardson, C. C. A Covalent Linkage between the Gene 5 DNA Polymerase of Bacteriophage T7 and Escherichia coli Thioredoxin, the Processivity Factor: FATE OF THIOREDOXIN DURING DNA SYNTHESIS *. *Journal of Biological Chemistry* **278**, 23762–23772 (2003).
9. Tabor, S., Huber, H. E. & Richardson, C. C. Escherichia coli thioredoxin confers processivity on the DNA polymerase activity of the gene 5 protein of bacteriophage T7. *J Biol Chem* **262**, 16212–16223 (1987).
10. Dangerfield, T. L., Kirmizialtin, S. & Johnson, K. A. Conformational dynamics during misincorporation and mismatch extension defined using a DNA polymerase with a fluorescent artificial amino acid. *Journal of Biological Chemistry* **298**, 101451 (2022).

11. Dangerfield, T. L., Kirmizialtin, S. & Johnson, K. A. Substrate specificity and proposed structure of the proofreading complex of T7 DNA polymerase. *Journal of Biological Chemistry* **298**, (2022).
12. Hoekstra, T. P. *et al.* Switching between Exonucleolysis and Replication by T7 DNA Polymerase Ensures High Fidelity. *Biophysical Journal* **112**, 575–583 (2017).
13. Loparo, J. J., Kulczyk, A. W., Richardson, C. C. & Van Oijen, A. M. Simultaneous single-molecule measurements of phage T7 replisome composition and function reveal the mechanism of polymerase exchange. *Proceedings of the National Academy of Sciences of the United States of America* **108**, 3584–3589 (2011).
14. Duderstadt, K. E. *et al.* Simultaneous Real-Time Imaging of Leading and Lagging Strand Synthesis Reveals the Coordination Dynamics of Single Replisomes. *Mol Cell* **64**, 1035–1047 (2016).
15. Geertsema, H. J., Kulczyk, A. W., Richardson, C. C. & van Oijen, A. M. Single-molecule studies of polymerase dynamics and stoichiometry at the bacteriophage T7 replication machinery. *Proc Natl Acad Sci U S A* **111**, 4073–4078 (2014).
16. Yeeles, J. T. P. & Marians, K. J. The Escherichia coli replisome is inherently DNA damage tolerant. *Science* **334**, 235–238 (2011).
17. Liao, Y., Li, Y., Schroeder, J. W., Simmons, L. A. & Biteen, J. S. Single-Molecule DNA Polymerase Dynamics at a Bacterial Replisome in Live Cells. *Biophysical Journal* **111**, 2562–2569 (2016).
18. Beattie, T. R. *et al.* Frequent exchange of the DNA polymerase during bacterial chromosome replication. *Elife* **6**, (2017).
19. Lewis, J. S. *et al.* Single-molecule visualization of fast polymerase turnover in the bacterial replisome. *Elife* **6**, (2017).
20. Spenkeliink, L. M. *et al.* Recycling of single-stranded DNA-binding protein by the bacterial replisome. *Nucleic Acids Res* **47**, 4111–4123 (2019).

21. Li, Y., Chen, Z., Matthews, L. A., Simmons, L. A. & Biteen, J. S. Dynamic Exchange of Two Essential DNA Polymerases during Replication and after Fork Arrest. *Biophysical Journal* **116**, 684–693 (2019).
22. Dubiel, K. *et al.* Development of a single-stranded DNA-binding protein fluorescent fusion toolbox. *NUCLEIC ACIDS RESEARCH* **48**, 6053–6067 (2020).
23. Lewis, J. S. *et al.* Tunability of DNA Polymerase Stability during Eukaryotic DNA Replication. *Mol Cell* **77**, 17-25.e5 (2020).
24. Kapadia, N. *et al.* Processive Activity of Replicative DNA Polymerases in the Replisome of Live Eukaryotic Cells. *Mol Cell* **80**, 114-126.e8 (2020).
25. Milstein, J. N. & Meiners, J.-C. On the role of DNA biomechanics in the regulation of gene expression. *Journal of The Royal Society Interface* **8**, 1673–1681 (2011).
26. Liedl, T., Högberg, B., Tytell, J., Ingber, D. E. & Shih, W. M. Self-assembly of three-dimensional prestressed tensegrity structures from DNA. *Nature Nanotech* **5**, 520–524 (2010).
27. Gupta, S., Gellert, M. & Yang, W. Mechanism of mismatch recognition revealed by human MutS β bound to unpaired DNA loops. *Nat Struct Mol Biol* **19**, 72–78 (2012).
28. Marians, K. J. Lesion Bypass and the Reactivation of Stalled Replication Forks. *Annual Review of Biochemistry* **87**, 217–238 (2018).
29. Ibarra, B. *et al.* Proofreading dynamics of a processive DNA polymerase. *The EMBO Journal* **28**, 2794–2802 (2009).
30. Smiley, R. D., Zhuang, Z., Benkovic, S. J. & Hammes, G. G. Single-molecule Investigation of T4 Bacteriophage DNA Polymerase Holoenzyme. *Biochemistry* **45**, 7990–7997 (2006).
31. Nathan A., T. *et al.* Single-Molecule Studies of Fork Dynamics of Escherichia coli DNA Replication. *Nat Struct Mol Biol* **15**, 170–176 (2008).
32. Cerron, F. *et al.* Replicative DNA polymerases promote active displacement of SSB proteins during lagging strand synthesis. *Nucleic Acids Research* **47**, 5723–5734 (2019).

REVIEWER COMMENTS

Reviewer #1 (Remarks to the Author):

The authors have made careful revisions to the manuscript and addressed our concerns. We recommend its publication in Nature Communications.

Reviewer #2 (Remarks to the Author):

The thoughtful response to our questions includes many key details that will significantly enhance this manuscript for both the general reader and the specialist. The results represent a significant advance in our understanding of polymerase dynamics.

Reviewer #3 (Remarks to the Author):

This manuscript does not present any new information and does not meet the standards for rigorous biochemical analysis. Details are given in the attached pdf.

In their response to the prior critique, the authors have present little new data, only arguments that do not adequately address the fundamental flaws of the paper.

1. The authors inaccurately present this work as characterization of polymerization at a replication fork. They have not described the template/primer system they used, and all evidence points to the use of a single template and primer in the reaction, not a replication fork which is much more difficult to assemble and study. To present this as studies at a replication fork is misleading and dishonest.
2. This paper has nothing new to offer. In response to the criticisms of their methods, the authors present a table comparing the estimates of the rates of polymerase dissociation from the DNA to rates published in 1993. Although this comparison may be used to justify the validity of their methods, their errors are much larger than estimates published in 1993. Most importantly, this comparison demonstrates that there is nothing novel in the current publication.
3. The authors justify the use of their method based on the scores of publications using these techniques in the past two decades. The methods were limited when first presented and those limitations have not been rectified in the intervening years. The methods have limited time resolution and are limited by the inability to measure single nucleotide incorporation events. The authors have not clearly stated the limits of resolution in their measurements of DNA polymerization. Similarly, artifacts due to the addition of the SNAP tag cannot be brushed aside just because everyone else does its without appropriate controls.
4. In response to the prior criticism regarding use of a fusion between the polymerase and the thioredoxin cofactor, the authors state now refer to a 2003 paper by Johnson and Richardson and they blur the line between a fusion and a covalent disulfide crosslink. While Richardson paper demonstrates that the crosslink positions the thioredoxin appropriately for activity, this does not address at all whether fusing the thioredoxin to the N- or C-terminus is effective.

The figure at the right shows the structure of T7 DNA polymerase (gray) with thioredoxin in cyan. The N- and C- termini of the polymerase are shown in blue and red spheres, respectively. Attaching thioredoxin to either the N- or the C-terminus is not tenable, even with a very long linker.	[redacted]
--	-------------------

5. This and other publications using single molecule methods have overstated the significance of DNA polymerase pausing. No one has shown that the pausing is not an artefact caused by the methods of analysis. In any event, the fraction of polymerase molecules in a paused state is insignificant. Therefore, the one novel contribution of these methods, as applied to DNA polymerization, is unimportant and not original with this paper.

6. There is little doubt that the tension on the DNA introduces artefacts by perturbing the structure of the enzyme. At the very least the authors need to make measurements as a function of force and extrapolate to zero force.

7. The authors' opening statements that there is very little known about DNA polymerization does not reflect the state of the field. The mechanistic detail now known about the polymerase kinetics beyond the limits of single molecule methods.

Response to Reviewers' Comments

Reviewer #1 (Remarks to the Author):

The authors have made careful revisions to the manuscript and addressed our concerns. We recommend its publication in Nature Communications.

We thank the reviewers again for their effort and careful evaluation of our manuscript.

Reviewer #2 (Remarks to the Author):

The thoughtful response to our questions includes many key details that will significantly enhance this manuscript for both the general reader and the specialist. The results represent a significant advance in our understanding of polymerase dynamics.

We thank the reviewers again for their input and constructive suggestions to improve our manuscript.

Reviewer #3 (Remarks to the Author):

This manuscript does not present any new information and does not meet the standards for rigorous biochemical analysis. Details are given in the attached pdf. In their response to the prior critique, the authors have present little new data, only arguments that do not adequately address the fundamental flaws of the paper.

To address the concerns of the reviewer we have taken great care to revise our manuscript and now include a separate **Extended Materials** document, describing the construction of the pKYB1 DNA and the preparation of the tagged SNAP-DNA polymerase, to provide the detailed information requested. In the revised manuscript, we also explicitly acknowledge the limitations of our resolution in detecting single-nucleotide events. We have made it clear that our study utilizes recombined SNAP-DNA polymerase, detailing the rationale behind this choice and its relevance to our research objectives. To address concerns regarding the structural integrity and functionality of our recombined DNA polymerase, we now include a comparison between the predicted structure of SNAP-DNA polymerase and experimentally determined T7 DNA polymerase structures. To validate the functionality of the SNAP-DNA polymerase, we conducted comparative analysis using two independent assays: real-time DNA primer extension assay and single-molecule assay. Our point-to-point responses below are in blue and changes in the text since last round of revision are in red.

1. The authors inaccurately present this work as characterization of polymerization at a replication fork. They have not described the template/primer system they used, and all evidence points to the use of a single template and primer in the reaction, not a replication fork which is much more difficult to assemble and study. To present this as studies at a replication fork is misleading and dishonest.

Indeed, in our study, we employed a DNA construct that mimics parts of the replication fork. The DNA construct is prepared with biotin labels at both ends, based on established protocols¹. See **Figure R1A** (also in **Figure 1A** of the manuscript, also in **Extended**

Materials for detailed procedures) of the schematics of our DNA construct. Our construct includes a 25 nt 5'-end overhang to enable exonucleolysis, thereby creating a ss/dsDNA junction for polymerization under controlled tension conditions (**Figure R1B**, also in **Extended Figure 1**).

We agree with the referee that '*replication fork*' doesn't describe our assay properly and we now corrected the terminology "*DNA replication fork*" throughout the manuscript, and made it clear in our **Method** section that our DNA construct mimicks part of a replication system.

Figure R1. Schematic of the DNA template (**Figure R1A**) and sequence of PKYB 1 construct (**Figure R1B**) used in the current study. The design of this DNA construct can be referred to publications¹and Method section in the manuscript. The green letter in panel B indicated biotinylated dATP used for bead tethering.

2. This paper has nothing new to offer. In response to the criticisms of their methods, the authors present a table comparing the estimates of the rates of polymerase dissociation from the DNA to rates published in 1993. Although this comparison may be used to justify the validity of their methods, their errors are much larger than estimates published in 1993. Most importantly, this comparison demonstrates that there is nothing novel in the current publication.

We respectfully disagree with the reviewer's assessment in this regard. The referenced polymerase dissociation rate of 0.2 s^{-1} from studies published in 1993 was actually suggested by the reviewer in previous comments as a benchmark. Our aim in comparing the dissociation rates obtained in our study ($0.75\text{-}0.9 \text{ s}^{-1}$) with those from both ensemble research (0.2 s^{-1}) as well as other single-molecule studies ($0.6\text{-}1.3 \text{ s}^{-1}$)^{2,3} was to validate the accuracy and reliability of our measurements and analytical methods. This comparison is crucial for demonstrating that our single-molecule approach, despite of using fluorescently labelled protein, aligns well with established data.

Furthermore, the measured dissociation rate ($0.75\text{-}0.9 \text{ s}^{-1}$) across various force ranges (20-50pN) demonstrate a clear understanding of the dissociation dynamics of DNA polymerase under different tensions, contrary to the assertion that this range of rates represent a large error as suggested by the reviewer. We acknowledge the reviewer's concern regarding the novelty of our publication. However, we emphasize that our study's novelty lies in the application of a correlative single-molecule methodology to explore the dynamics of polymerase exchange in a more detailed and nuanced manner than previously possible.

3. The authors justify the use of their method based on the scores of publications using these techniques in the past two decades. The methods were limited when first presented and those limitations have not been rectified in the intervening years. The methods have limited time resolution and are limited by the inability to measure single nucleotide incorporation events. The authors have not clearly stated the limits of resolution in their

measurements of DNA polymerization. Similarly, artifacts due to the addition of the SNAP tag cannot be brushed aside just because everyone else does its without appropriate controls.

We acknowledge the limitations mentioned, specifically the temporal resolution constraints and the challenge in detecting single-nucleotide incorporation events directly. Our experimental setup, combining optical tweezers with fluorescence microscopy, offers a temporal resolution ($\sim 0.3\text{s}/\text{scan}$) that, while not sufficient to resolve individual nucleotide incorporations, is highly effective in capturing the rapid exchange dynamics of DNA polymerase and its overall activity patterns. To address the reviewer's comments, we have added the following text to the manuscript to clarify the resolution limits of our experimental setup:

"The temporal resolution of our correlative optical tweezers-fluorescence microscopy setup is determined by the confocal scanning rate and the signal acquisition time required for reliable fluorescence detection. While this resolution does not allow for direct observation of single nucleotide incorporation events, it is optimized to capture the rapid exchange dynamics of DNA polymerase and polymerase activity patterns during replication, complementing higher-resolution techniques."

To address the concerns about the effect of SNAP tag fusion and subsequent fluorescence labelling, we first investigate the structural integrity of our recombinant DNA polymerase by employing the RoseTTAFold Modeling Method⁴ to predict the structure of our utilized SNAP-DNA polymerase. Aligning this model with the experimentally determined structure of T7 DNA polymerase (PDB:1T7P) and a DNA primer/template complex allowed us to verify that the structural integrity and active site functionality is expected not to be impacted. This analysis, presented in **Figure R2** and **Extended Figure 3**, shows that the SNAP-tag is out of the way and should minimally impact the polymerase's functionality.

Figure R2. Structure and Alignment of Predicted SNAP-DNA Polymerase with T7 DNA Polymerase.
(A) Predicted structure of SNAP-DNA polymerase using the RoseTTAFold method⁵, aligned with the

experimentally determined T7 DNA polymerase (PDB:1T7P)⁴ to model the complex with DNA primer/template. Color: SNAP-tag in magenta, N-terminal with flexible GS-linker in red, and gp5 protein in cyan. (B) Comparison of the RoseTTAFold-predicted SNAP-DNA polymerase structure with T7 DNA polymerase (PDB:1T7P), demonstrating preserved structure and active site for DNA binding. The experimental T7 polymerase is in green, with trxA indicated in light green. (C) Close-up of the N-terminal with a GS-linker and SNAP-tag, illustrating the tag's distance from the active site, suggesting minimal impact on binding and activity. (D) Zoomed-in view of the trxA-binding domain, showing its accessibility for trxA interaction.

We also conducted control experiments, using real-time DNA primer extension assay⁶ and single-molecule assays³, showing that our SNAP-tagged polymerase has comparable activity to commercial T7 DNA polymerase (NEB, # M0274L) (**Extended Materials, Figure R3**, also in **Extended Figure 4**). In particular, using a real-time DNA primer extension assay, we compared DNA polymerase activity across a range of enzyme concentrations. The assay measured the effect of nucleotide incorporation on the quenching of the intensity of 5' fluorophores at various DNAP concentrations. Our results demonstrate that both the commercial T7 DNA polymerase (**Figure R3A**, shown with a gradient of blue) and our SNAP-tagged polymerase (**Figure R3A**, shown with a gradient of green) displayed similar activities under our measured concentrations.

Figure R3. SNAP-DNAP exhibits activity comparable to commercial DNAP. (A) Analysis of DNA polymerase activity using a real-time DNA primer extension assay reveals comparable performance between commercial T7 DNA polymerase (illustrated with a gradient of blue) and SNAP-DNAP pre-mixed with trxA at a 1:4 ratio (shown with a gradient of green), across a range of enzyme concentrations (2.3nM, 4.7nM and 9.4nM). The assay measures the quenching of a 5' fluorophore's intensity by nucleotide incorporation in the top strand. Data were plotted with three independent measurements. (B) Polymerase activity was quantified by analyzing the initial linear phase of the fluorescence intensity decrease (first 3 minutes). Values and errors (sem) were derived from three independent experiments. Furthermore, we compare the DNAP activity using single-molecule assay (C and D). Comparative Analysis of Polymerization and Exonucleolysis Activity Between

Commercial T7 DNA Polymerase (Control, panel C) conducted in the same lab using same analysis method, adapted from ref³ and Modified DNA Polymerase (Experiment, panels D).

Further, at the single-molecule level, we compared the activity of both polymerases under various tensions. This comparison showed that the SNAP-tagged polymerase's (**Figure R3B**) kinetics closely match those of the published data (see **Figure R3A**)³. These findings indicate that the SNAP tag does not significantly impact the enzyme's activity or DNA interaction. As suggested, we have also refined the manuscript to explicitly state that our findings are based on a recombinant DNA polymerase with a SNAP-tag. The manuscript now includes the following statement:

"In this study, we used a recombinant DNA polymerase with a SNAP-tag for protein labelling. Future research should consider direct labelling methods for the polymerase, although attempts to do so were not successful in our case."

4. In response to the prior criticism regarding use of a fusion between the polymerase and the thioredoxin cofactor, the authors state now refer to a 2003 paper by Johnson and Richardson and they blur the line between a fusion and a covalent disulfide crosslink. While Richardson paper demonstrates that the crosslink positions the thioredoxin appropriately for activity, this does not address at all whether fusing the thioredoxin to the N- or C-terminus is effective.

The figure at the right shows the structure of T7 DNA polymerase (gray) with thioredoxin in cyan. The N- and C- termini of the polymerase are shown in blue and red spheres, respectively. Attaching thioredoxin to either the N- or the C-terminus is not tenable, even with a very long linker.	[redacted]
--	-------------------

We appreciate the reviewers' insightful comments and the opportunity to clarify our experimental approach. Upon re-examination of our protein sequence, laboratory documents, and experimental data, we recognized an oversight in our initial design. Our prior description mistakenly suggested the successful expression of a thioredoxin fused directly to the T7 DNA polymerase. However, detailed examination of our protein sequence and laboratory documentation revealed that attempts to create a trxA-gp5 fusion protein were unsuccessful. This was due to the presence of stop codons immediately following the gp5 gene, which led to premature translation termination. This was conclusively identified through the use of the ExPASy Translation Tool for predicting protein translation (**Figure R4A**) and confirmed by SDS-PAGE. The observed molecular weight was consistent with the absence of a trxA-tagged component, as shown in **Figure R4B** and detailed in the **Extended Materials**.

Figure R4: Analysis of SNAP-DNA Polymerase Translation and Purification. (A) Protein translation prediction for the pETDuet-1_SNAP-DNA polymerase plasmid, using the ExPASy Translation Tool from SIB Swiss Institute of Bioinformatics. Results show that *trx*A-tag fusion is disrupted by stop codons post-*gp5* gene, with only the DNA polymerase's open-reading-frame successfully expressed (highlighted in red). (B) SDS-PAGE gel electrophoresis confirmed the purified product as solely SNAP-DNA polymerase, with the *trx* component absent.

Our initial intent was inspired by the potential to enhance T7 DNA polymerase's processivity through thioredoxin fusion. Addressing the reviewer's concern, we acknowledge the practical challenges of effectively attaching thioredoxin to either the N- or C-terminus of the polymerase, even with the use of a lengthy linker. In our study, the addition of an excess amount of free thioredoxin—a common practice for working with T7 DNA polymerase^{1,13}—resulted in normal enzymatic activity even though the fusion was not successful. Thus, this methodological clarification does not alter the interpretation of our data, or the conclusions drawn from our study and in fact might be a blessing in disguise. We have amended the **Methods** section of our manuscript to accurately reflect these findings and clarify the experimental design. We regret any confusion our initial submission may have caused and are grateful for the chance to correct this error.

5. This and other publications using single molecule methods have overstated the significance of DNA polymerase pausing. No one has shown that the pausing is not an artefact caused by the methods of analysis. In any event, the fraction of polymerase molecules in a paused state is insignificant. Therefore, the one novel contribution of these methods, as applied to DNA polymerization, is unimportant and not original with this paper.

We respectfully disagree with the assertion that the significance of DNA polymerase pausing is merely an artifact of analysis methods or is insignificant. Firstly, we employed two independent methods (mechanical force and fluorescence microscopy) to analyze and interpret our data, ensuring that the observed pausing events are not artifacts of a particular analysis technique. See example pausing events detected in our study, from mechanical measurement (**Figure R5A, red arrow**) and fluorescence measurements (**Figure R5B, red arrow**). This multi-faceted approach strengthens the validity of our findings, demonstrating that pausing is an inherent feature of DNA polymerase dynamics rather than a methodological artifact. Contrary to the reviewer's assertion, our data indicate that pauses are not merely stochastic interruptions but are integral to the polymerase's function, possibly allowing for exchange of DNA polymerase and processivity regulation (**Figure R5C and R5D**). This aligns with emerging evidence suggesting that transient pausing can influence enzymatic activity and genome stability.

Figure R5. Representative Pausing Events and Integrated Pause Replication Process. In the current study, pausing events are captured via two independent methods: optical tweezer mechanical measurements (Figure R5A) and confocal microscopy imaging (Figure R5B). Correlation of these datasets identifies replication segments with pauses, highlighting a prevalence of single-type burst segments (Figure R5C) and transitional-type burst events (Figure R5D).

Furthermore, it is crucial to recognize that DNA polymerase pausing has been documented across various studies with various methods, attributed to a multitude of factors such as DNA secondary structures⁷, the presence of DNA lesions^{8,9}, specific sequences¹⁰ and replication-transcription interaction¹¹. In our study, we extend the understanding of DNA polymerase dynamics by suggesting that dissociation events could contribute to observed pausing. Contrary to the assertion that the pausing is "insignificant" and our contributions "unimportant and not original," we argue that elucidating the underpinnings of polymerase behavior, including pausing, enriches our comprehension of the replication machinery's adaptability and error-correction mechanisms.

Finally, our work extends beyond merely identifying pausing events. We offer new insights into the context and implications of these pauses, particularly in relation to the enzyme's overall activity and replication fidelity. This is supported by a detailed analysis of pause frequency, duration, and correlation with enzymatic activity changes, providing a more comprehensive picture of DNA polymerase behavior than previously reported.

6. There is little doubt that the tension on the DNA introduces artefacts by perturbing the structure of the enzyme. At the very least the authors need to make measurements as a function of force and extrapolate to zero force.

The impact of force on DNA polymerase activity has been studied before and discussed in quite some detail^{2,3}. But as suggested, we conducted additional experiments across a range of applied forces, including analysis at low tensions at 10pN to assess the potential impact of applied tension on the DNA structure and, consequently, on the observed behavior of the DNA polymerase (see Figure R2B, also in **Extended Figure 4**). That said, for exonuclease (exo) activity, data has shown that this reaction appears largely independent of the applied

force. This independence from force underscores a fundamental aspect of DNA polymerase's function—its ability to conduct proofreading without being significantly influenced by the mechanical state of the DNA. Such a characteristic is essential for the enzyme's role in maintaining genomic integrity across various physiological conditions.

Moreover, it is not very likely that DNA polymerase inside cells/bacteria will process DNA that is under zero tension, given the many constraints on the DNA inside the crowded interior and it being continuously processed by many proteins. Thus extrapolating to zero force will not provide more insight than the force range that we considered.

7. The authors' opening statements that there is very little known about DNA polymerization does not reflect the state of the field. The mechanistic detail now known about the polymerase kinetics beyond the limits of single molecule methods.

It appears there has been a misunderstanding about the phrasing of our manuscript's introduction. Our manuscript states, "*Despite extensive studies on DNA replication, the exchange mechanisms of DNA polymerase at the replication fork (now changed to 'during replication') remain unclear.*" Our intent was not to suggest a general lack of knowledge about DNA polymerization mechanisms, which indeed are well-characterized through extensive biochemical, structural, and mechanistic studies. Instead, our focus was specifically on the exchange mechanisms of DNA polymerase at the replication fork, an area that remains less understood, especially in the context of dynamic, real-time processes observable through single-molecule techniques.

This distinction is crucial, as the rapid and autonomous exchange of DNA polymerase at the replication fork, as observed in our study, underscores complex regulatory mechanisms that govern replication fidelity and efficiency in a cellular context. Our findings, as detailed in both the manuscript and the response to the first round of revisions, highlight these dynamic processes, contributing to the broader discourse on DNA replication mechanics.

References

1. Candelli, A. *et al.* A toolbox for generating single-stranded DNA in optical tweezers experiments. *Biopolymers* **99**, 611–620 (2013).
2. Wuite, G. J. L., Smith, S. B., Young, M., Keller, D. & Bustamante, C. Single-molecule studies of the effect of template tension on T7 DNA polymerase activity. *Nature* **404**, 103–106 (2000).
3. Hoekstra, T. P. *et al.* Switching between Exonucleolysis and Replication by T7 DNA Polymerase Ensures High Fidelity. *Biophysical Journal* **112**, 575–583 (2017).
4. Doublé, S., Tabor, S., Long, A. M., Richardson, C. C. & Ellenberger, T. Crystal structure of a bacteriophage T7 DNA replication complex at 2.2 Å resolution. *Nature* **391**, 251 (1998).
5. Baek, M. *et al.* Accurate prediction of protein structures and interactions using a three-track neural network. *Science* **373**, 871–876 (2021).

6. Toste Rêgo, A., Holding, A. N., Kent, H. & Lamers, M. H. Architecture of the Pol III–clamp–exonuclease complex reveals key roles of the exonuclease subunit in processive DNA synthesis and repair. *EMBO J* **32**, 1334–1343 (2013).
7. Kuzminov, A. Inhibition of DNA synthesis facilitates expansion of low-complexity repeats. *BioEssays* **35**, 306–313 (2013).
8. Lehmann, A. R. *et al.* Translesion synthesis: Y-family polymerases and the polymerase switch. *DNA Repair* **6**, 891–899 (2007).
9. Courcelle, J., Donaldson, J. R., Chow, K.-H. & Courcelle, C. T. DNA Damage-Induced Replication Fork Regression and Processing in Escherichia coli. *Science* **299**, 1064–1067 (2003).
10. Lerner, L. K. & Sale, J. E. Replication of G Quadruplex DNA. *Genes* **10**, 95 (2019).
11. Azvolinsky, A., Giresi, P. G., Lieb, J. D. & Zakian, V. A. Highly Transcribed RNA Polymerase II Genes Are Impediments to Replication Fork Progression in Saccharomyces cerevisiae. *Molecular Cell* **34**, 722–734 (2009).

REVIEWERS' COMMENTS

Reviewer #3 (Remarks to the Author):

See attached pdf

On this third round of review, the authors have finally admitted that they are not studying DNA polymerization at a replication fork but rather have studied a simple primer/template system that has been studied extensively for decades. In addition, they now agree that it is untenable to attach the thioredoxin accessory protein to the N- or C-terminus of the polymerase—they now claim that the studies were done by adding excess thioredoxin. The authors have provided new computational analysis to suggest that the attachment of the SNAP tag to the N-terminus via a long linker may not alter the structure of the enzyme, but this attachment in the exonuclease domain could affect enzyme dynamics to alter the exonuclease reaction rates. The authors have described that their sampling rate is only 3 per second. Given a polymerase rate of 500/s they are only measuring DNA polymerization with a resolution of 170 nucleotides per time point. Moreover, the compliance in the assay (flexibility of the DNA) dampens out any fast processes related to single nucleotide incorporation events. There are still unresolved issues with this manuscript, the most serious of which is the overstated claims of significance of their data toward understanding DNA polymerase kinetics and mechanisms.

The state-of-the-art in studying DNA replication is based on the ability to measure reactions using multiple signals on a millisecond timescale during a single nucleotide incorporation event (1-5). As is common for single molecule papers, the authors have ignored the prior literature in this field and thereby claim to have made an original discovery. The authors claim to be advancing our understanding of DNA replication while being totally unaware of the preceding literature and the current state-of-the-art. In general, the authors' lack of understanding and dismissal of the prior literature has led to many of the weaknesses of the prior and current versions of this manuscript and prior publications using these methods. The premise of the paper is that rate of enzyme dissociation DNA is not known. The rate of dissociation has been measured for every DNA polymerase that has been studied in detail (Klenow, T7 DNA pol, HIV RT, human mitochondrial DNA pol, pol beta, Pol C).

On the millisecond timescale, pausing events that affect the authors' single molecule measurements do not affect single turnover kinetic studies. Such pausing can be explained, in part, by the low concentration of enzyme relative to the K_d for DNA binding in the single molecule assays. In addition, although single molecule experiments of the type used here have led to the proposal that pausing could be due to secondary structure. In contrast, detailed single turnover experiments have quantified the kinetics of reading through DNA and RNA hairpins in with single nucleotide resolution (6, 7).

In the revised manuscript, the authors have provided new polymerase activity measurements in an attempt to show that the SNAP tag does not alter activity (Extended Data Figure 4). The authors' interpretation of the results is incorrect; the experiment measures rate-limiting DNA dissociation from the enzyme, not polymerization. Although not a steady-state experiment per se, it corresponds to a full reaction progress steady-state turnover experiment. There are several limitations to their analysis.

- a. Fitting the initial rate and reporting RFU/s is meaningless.

- b. NEB sells T7 DNA polymerase in a solution of 10,000 units/ml. The authors must describe how they measured the concentration of enzyme for both enzyme samples and how they measured the fraction of enzyme that is active. The data presented for the two enzymes are suspiciously closer than one would expect for two different preparations of the enzyme made in two different labs. The rates depend on the enzyme concentrations and so the methods used to measure the concentration must be documented.
- c. The authors state that 70 nM labeled DNA was used. In the original description of the method which they reference, Rego et al., added 60 nM labeled DNA and 1 μ M unlabeled DNA. The authors of the current manuscript must have also added unlabeled DNA; otherwise, their results would require a DNA dissociation rate constant of 0.03 s⁻¹. The addition of 1 μ M DNA needs to be added to the methods section if that is the case.
- d. The experiment in Extended Data Figure 4 can be modeled as shown below in Figure 1. Polymerization to incorporate 12 nucleotides is fast (reaching 90% completion in 40 ms) and orders of magnitude faster than DNA release. Using estimates of the half-life of the reaction for the authors data and assuming a DNA concentration of 1070 nM gives a dissociation rate constant $k_3 = 0.45$ s⁻¹. The data in Extended Data Figure 4 could give an accurate estimate of the DNA release rate if fitted properly.
- e. Fitting by numerical integration of the rate equations will give the DNA dissociation rate. In fact, this assay provides a simpler and more accurate measurement of DNA dissociation rate than the authors single molecule assay. Figure 1 shows the modeling of the reaction assuming 1.07 μ M DNA concentration as described in the original paper detailing this method (Rego et al.2013). The authors' data is well represented with a DNA dissociation rate of 0.45 s⁻¹ with the rate constants given in the pathway below leading to kinetic traces as shown in Figure 1.

Each incorporation occurs at a net rate of 300 s⁻¹ at 25 μ M dNTP in the experiment. The reaction to incorporate 12 nucleotides proceeds at a net rate of 25 s⁻¹, which is much faster than DNA release, the rate-limiting step in the measurement.

Figure 1. Computer simulations of the reaction were generated at 2.4, 4.7, and 9.4 nM enzyme with 1 μ M DNA (70 nM labeled DNA) and the rate constants shown above. Confidence contour analysis based on this simulation demonstrate that the only parameter defined by this experiment is the DNA dissociation rate (assumed to be the same for substrate and product). This figure reproduces the authors' Extended Data Figure 4.

Figure 1A. Simulation to mimic the authors' data.

Figure B at the right shows the time course if 70 nM DNA was used but the DNA dissociation rate was 0.45 s^{-1} . Alternatively, if indeed only 70 nM DNA was used, then the dissociation rate would need to be 0.03 s^{-1} to account for the data. The most reasonable interpretation is that the authors overlooked the addition of $1 \mu\text{M}$ unlabeled DNA in the writeup of the methods.

Figure 1B. Simulation with only 70 nM DNA

Summary

The final conclusion of the paper, if we are to accept the results, is that the authors have provided an independent measurement of the enzyme-DNA dissociation rate which is 3-fold faster than prior measurements performed at different salt concentration (50 versus 100 mM NaCl) (1, 2). A threefold difference in rate is not noteworthy. Moreover, the authors fail to recognize prior work by Richardson's lab to show that T7 DNA polymerases exchange at a true replication fork (8). These prior results using a true replication fork further negate the stated novelty of the current manuscript.

Editorial Note:

The authors still refer to the use of thioredoxin fused to the polymerase in the legend to Figure 1, and on line 175 of the methods supplement the authors describe a long linker between the polymerase and thioredoxin in the gene sequence. What did the authors really do in these studies? These inconsistencies shed doubt on the belated statement that they abandoned the thioredoxin fusion protein and added thioredoxin separately.

1. Patel, S. S., Wong, I., and Johnson, K. A. (1991) Pre-Steady-State Kinetic-Analysis of Processive DNA-Replication Including Complete Characterization of An Exonuclease-Deficient Mutant *Biochemistry* **30**, 511-525, ISI:A1991ET38100029
2. Dangerfield, T. L., and Johnson, K. A. (2021) Conformational dynamics during high-fidelity DNA replication and translocation defined using a DNA polymerase with a fluorescent artificial amino acid *Journal of Biological Chemistry* **296**, 100143-100161 10.1074/jbc.RA120.016617

3. Dangerfield, T. L., Kirmizialtin, S., and Johnson, K. A. (2022) Substrate specificity and proposed structure of the proofreading complex of T7 DNA polymerase *J Biol Chem* **298**, 101627 10.1016/j.jbc.2022.101627
4. Dangerfield, T. L., and Johnson, K. A. (2023) Kinetics of DNA strand transfer between polymerase and proofreading exonuclease active sites regulates error correction during high-fidelity replication *Journal of Biological Chemistry* **299**, 102744 10.1016/j.jbc.2022.102744
5. Dangerfield, T. L., and Johnson, K. A. (2023) Design and interpretation of experiments to establish enzyme pathway and define the role of conformational changes in enzyme specificity *Methods in enzymology* **685**, 461-492 10.1016/bs.mie.2023.03.018
6. Suo, Z., and Johnson, K. A. (1997) Effect of RNA secondary structure on the kinetics of DNA synthesis catalyzed by HIV-1 reverse transcriptase *Biochemistry* **36**, 12459-12467, PM:9376350
7. Suo, Z., and Johnson, K. A. (1998) DNA secondary structure effects on DNA synthesis catalyzed by HIV-1 reverse transcriptase *J Biol Chem* **273**, 27259-27267, PM:9765249
8. Johnson, D. E., Takahashi, M., Hamdan, S. M., Lee, S. J., and Richardson, C. C. (2007) Exchange of DNA polymerases at the replication fork of bacteriophage T7 *Proc Natl Acad Sci U S A* **104**, 5312-5317 10.1073/pnas.0701062104

Reviewer #4 (Remarks to the Author):

In this manuscript, the authors have employed correlative fluorescence microscopy and force spectroscopy to elucidate the dynamics of the T7 DNA polymerase gp5 during its exonuclease and polymerase activities. Their study uncovers a rapid polymerase exchange phenomenon in the presence of the processivity factor thioredoxin, independent of the complete replisome assembly. Additionally, they report a 'memory effect' influencing polymerase exchange, which they attribute to the state of the single-stranded/double-stranded DNA junction.

The manuscript has undergone prior review by three referees, with the first two expressing satisfaction upon revision. The third reviewer, however, persisted with concerns. Initially, the critique centered on the potential non-representative nature of the gp5-thioredoxin-SNAP-tag fusion and the possible influence of applied force on DNA affecting the gp5 activity. The authors have since addressed these points satisfactorily in their revision.

Subsequently, the third reviewer raised issues regarding the novelty of the findings, suggesting that polymerase exchange has been observed at authentic replication forks involving helicase-mediated DNA unwinding. I find the insights provided by the current work, particularly the independence of polymerase exchange from helicase interaction and the noted 'memory effect,' to be significant contributions. The third reviewer's insistence on single-nucleotide resolution overlooks the value of single-molecule visualization techniques in revealing enzymatic reaction heterogeneities and kinetics, despite lower spatial resolution.

The thorough critique from the third referee has led to an important realization that the gp5 construct used lacked a thioredoxin fusion. But, it seems that the authors had also included gp5 in their experiments. Overall, I feel that the authors have addressed the comments from the third reviewer, enhancing the quality of the manuscript.

I would advise the authors to consider the remaining remarks of the third reviewer, particularly pertaining to Extended Data Figure 4. The concerns about novelty should not overshadow the merits of the study. Once these final points are addressed, I advocate for the manuscript's publication in Nature Communications.

Reviewer #3 (Remarks to the Author):

[...]

In the revised manuscript, the authors have provided new polymerase activity measurements in an attempt to show that the SNAP tag does not alter activity (Extended Data Figure 4). The authors' interpretation of the results is incorrect; the experiment measures rate-limiting DNA dissociation from the enzyme, not polymerization. Although not a steady-state experiment per se, it corresponds to a full reaction progress steady-state turnover experiment. There are several limitations to their analysis.

We respectfully disagree with this assessment of the reviewer as is described below.

- a. Fitting the initial rate and reporting RFU/s is meaningless.

Reporting initial rates (RFU/s) is widely accepted¹⁻⁴ in analyzing polymerase activity, especially with an exonuclease domain. While RFU/s may not be the most informative metric, it serves as an internally consistent measure for comparing relative activities within the same experimental system and a reasonable approximation for comparing relative polymerase activities between different enzyme preparations, as our primary goal was to evaluate the impact of the SNAP tag on polymerase activity.

- b. NEB sells T7 DNA polymerase in a solution of 10,000 units/ml. The authors must describe how they measured the concentration of enzyme for both enzyme samples and how they measured the fraction of enzyme that is active. The data presented for the two enzymes are suspiciously closer than one would expect for two different preparations of the enzyme made in two different labs. The rates depend on the enzyme concentrations and so the methods used to measure the concentration must be documented.

The protein concentration for both commercial and SNAP-DNAp was determined using a Bradford assay, as clarified in the Supplementary Methods.

- c. The authors state that 70 nM labeled DNA was used. In the original description of the method which they reference, Rego et al., added 60 nM labeled DNA and 1 μ M unlabeled DNA. The authors of the current manuscript must have also added unlabeled DNA; otherwise, their results would require a DNA dissociation rate constant of 0.03 s^{-1} . The addition of 1 μ M DNA needs to be added to the methods section if that is the case.

In our experimental setup, we used 70 nM labeled DNA without the addition of 1 μ M unlabeled DNA, as stated in our methods section. The reason for this is that the exonuclease activity of the polymerase can degrade the polymerization product, leading to an apparent lower rate of fluorescence signal change compared to the scenario without exonuclease activity.

d. The experiment in Extended Data Figure 4 can be modeled as shown below in Figure 1. Polymerization to incorporate 12 nucleotides is fast (reaching 90% completion in 40 ms) and orders of magnitude faster than DNA release. Using estimates of the half-life of the reaction for the authors data and assuming a DNA concentration of 1070 nM gives a dissociation rate constant $k_3 = 0.45 \text{ s}^{-1}$. The data in Extended Data Figure 4 could give an accurate estimate of the DNA release rate if fitted properly.

While the reviewers' kinetic model and simulations provide valuable insights, we believe incorporating exonuclease activity is crucial for accurately describing the observed fluorescence signal changes in our system. See answer below the figures of the reviewer

e. Fitting by numerical integration of the rate equations will give the DNA dissociation rate. In fact, this assay provides a simpler and more accurate measurement of DNA dissociation rate than the authors single molecule assay. Figure 1 shows the modeling of the reaction assuming $1.07 \mu\text{M}$ DNA concentration as described in the original paper detailing this method (Rego et al.2013). The authors' data is well represented with a DNA dissociation rate of 0.45 s^{-1} with the rate constants given in the pathway below leading to kinetic traces as shown in Figure 1.

Each incorporation occurs at a net rate of 300 s^{-1} at $25 \mu\text{M}$ dNTP in the experiment. The reaction to incorporate 12 nucleotides proceeds at a net rate of 25 s^{-1} , which is much faster than DNA release, the rate-limiting step in the measurement.

Figure 1B at the right shows the time course if 70 nM DNA was used but the DNA dissociation rate was 0.45 s^{-1} . Alternatively, if indeed only 70 nM DNA was used, then the dissociation rate would need to be 0.03 s^{-1} to account for the data. The most reasonable interpretation is that the authors overlooked the addition of $1 \mu\text{M}$ unlabeled DNA in the writeup of the methods.

Figure 1B. Simulation with only 70 nM DNA

The reviewers' kinetic model assumes that the observed fluorescence signal change is solely governed by the DNA dissociation rate from the enzyme-product complex. However, this assumption neglects the potential contribution of the exonuclease activity, which can degrade the polymerized product, leading to an increase in the fluorescence signal, as observed at higher polymerase concentrations (**Figure R1A**). Although **Extended Figure 4** (now revised as **Supplementary Figure 7**) presents polymerase activity at relatively low concentrations (2.4, 4.7, and 9.4 nM), T7 DNA polymerase can potentially perform exonuclease activity, removing nucleotides and degrading the enzyme-product complex. Including the exonuclease activity is thus needed but it makes the kinetic model more complex, see **Figure R1B**^{5,6}. Hence, we chose to report the initial rates (RFU/s) as an indication of polymerase activity, rather than fitting the data to a potentially oversimplified kinetic model that neglects exonuclease activity as suggested by the reviewer. And with that approach we can indeed show that the SNAP tag does not alter activity.

Figure R1. (A) Real-time DNA primer extension assay demonstrates that T7 DNA polymerase exhibits both polymerization and exonuclease activity at high concentrations (37.5 nM). Although at the lower concentration of 9.4 nM used in **Extended Figure 4 (now revised as Supplementary Figure 7)**, the reaction is primarily dominated by polymerization, the enzyme can intrinsically perform exonuclease activity. Note that the addition of polymerase to DNA initiates an immediate reaction, often resulting in the loss of initial data points. (B) An example of established kinetic pathway of T7 DNA polymerase, considering both the polymerization and exonuclease activity. The states of the DNA-enzyme complex are represented as: E, free enzyme in solution; E_p, x , DNA bound to the polymerase or exonuclease active sites; D_n , DNA primer with length n ; dNTP, deoxynucleotide; and PP_i , pyrophosphate. This kinetic model is derived from prior studies⁵.

[...]

References

1. Ma, C., Liu, H., Wang, J., Jin, S. & Wang, K. Label-free molecular beacon for real-time monitoring of DNA polymerase activity. *Anal Bioanal Chem* **408**, 3275–3280 (2016).
2. Dorjsuren, D. *et al.* A real-time fluorescence method for enzymatic characterization of specialized human DNA polymerases. *Nucleic Acids Research* **37**, e128 (2009).
3. Toste Rêgo, A., Holding, A. N., Kent, H. & Lamers, M. H. Architecture of the Pol III–clamp–exonuclease complex reveals key roles of the exonuclease subunit in processive DNA synthesis and repair. *EMBO J* **32**, 1334–1343 (2013).
4. Chengalroyen, M. D. *et al.* DNA-Dependent Binding of Nargenicin to DnaE1 Inhibits Replication in *Mycobacterium tuberculosis*. *ACS Infect. Dis.* **8**, 612–625 (2022).
5. Wuite, G. J. L., Smith, S. B., Young, M., Keller, D. & Bustamante, C. Single-molecule studies of the effect of template tension on T7 DNA polymerase activity. *Nature* **404**, 103–106 (2000).
6. Hoekstra, T. P. *et al.* Switching between Exonucleolysis and Replication by T7 DNA Polymerase Ensures High Fidelity. *Biophysical Journal* **112**, 575–583 (2017).

Editorial Note:

The authors still refer to the use of thioredoxin fused to the polymerase in the legend to Figure 1, and on line 175 of the methods supplement the authors describe a long linker between the polymerase and thioredoxin in the gene sequence. What did the authors really do in these studies? These inconsistencies shed doubt on the belated statement that they abandoned the thioredoxin fusion protein and added thioredoxin separately.

We apologize for the confusion. The references in the legend to **Figure 1** were errors from an oversight during the last revision and have now been corrected.

Initially, we attempted to engineer a fusion of thioredoxin with the polymerase, including a long linker between the polymerase and thioredoxin in the gene sequence (on line 175 of the **Extended Materials** of prior version). However, this approach failed due to the presence of stop codons immediately following the gp5 gene, which prevented successful translation of the trxA fusion. This issue was conclusively identified through translation prediction using the ExPasy Translation Tool (SIB Swiss Institute of Bioinformatics) and confirmed by molecular weight analysis with SDS-PAGE, as detailed in **Supplementary Figure 5**. Consequently, no thioredoxin fusion protein was successfully expressed.

We have now corrected the **Supplementary Information** and figure legends to accurately reflect that thioredoxin was added separately to the experiments, and not as a part of a fusion protein with the polymerase.

Reviewer #4 (Remarks to the Author):

In this manuscript, the authors have employed correlative fluorescence microscopy and force spectroscopy to elucidate the dynamics of the T7 DNA polymerase gp5 during its exonuclease and polymerase activities. Their study uncovers a rapid polymerase exchange phenomenon in the presence of the processivity factor thioredoxin, independent of the complete replisome assembly. Additionally, they report a 'memory effect' influencing polymerase exchange, which they attribute to the state of the single-stranded/double-stranded DNA junction.

The manuscript has undergone prior review by three referees, with the first two expressing satisfaction upon revision. The third reviewer, however, persisted with concerns. Initially, the critique centered on the potential non-representative nature of the gp5-thioredoxin-SNAP-tag fusion and the possible influence of applied force on DNA affecting the gp5 activity. The authors have since addressed these points satisfactorily in their revision.

Subsequently, the third reviewer raised issues regarding the novelty of the findings, suggesting that polymerase exchange has been observed at authentic replication forks involving helicase-mediated DNA unwinding. I find the insights provided by the current work, particularly the independence of polymerase exchange from helicase interaction and the noted 'memory effect,' to be significant contributions. The third reviewer's insistence on single-nucleotide resolution overlooks the value of single-molecule visualization techniques in revealing enzymatic reaction heterogeneities and kinetics, despite lower spatial resolution.

The thorough critique from the third referee has led to an important realization that the gp5 construct used lacked a thioredoxin fusion. But, it seems that the authors had also included gp5 in their experiments. Overall, I feel that the authors have addressed the comments from the third reviewer, enhancing the quality of the

manuscript.

I would advise the authors to consider the remaining remarks of the third reviewer, particularly pertaining to Extended Data Figure 4. The concerns about novelty should not overshadow the merits of the study. Once these final points are addressed, I advocate for the manuscript's publication in Nature Communications.

We deeply appreciate the Reviewer #4 assessment of our manuscript and as requested have addressed the comments from Reviewer #3 regarding **Extended Data Figure 4** (now revised as **Supplementary Figure 7**). For details on these revisions, please refer to the explanations provided above.